# Neuro-Inspired Information-Theoretic Hierarchical Perception for Multimodal Learning

**Xiongye Xiao**[1*†], **Gengshuo Liu**[1*], **Gaurav Gupta**[1], **Defu Cao**[1], **Shixuan Li**[1], **Yaxing Li**[1],
**Tianqing Fang**[2], **Mingxi Cheng**[1] & **Paul Bogdan**[1†]

[1]University of Southern California, Los Angeles, CA 90089, USA
[2]Hong Kong University of Science and Technology, Hong Kong, China

## Abstract

Integrating and processing information from various sources or modalities are critical for obtaining a comprehensive and accurate perception of the real world in autonomous systems and cyber-physical systems. Drawing inspiration from neuroscience, we develop the Information-Theoretic Hierarchical Perception (ITHP) model, which utilizes the concept of information bottleneck. Different from most traditional fusion models that incorporate all modalities identically in neural networks, our model designates a prime modality and regards the remaining modalities as detectors in the information pathway, serving to distill the flow of information. Our proposed perception model focuses on constructing an effective and compact information flow by achieving a balance between the minimization of mutual information between the latent state and the input modal state, and the maximization of mutual information between the latent states and the remaining modal states. This approach leads to compact latent state representations that retain relevant information while minimizing redundancy, thereby substantially enhancing the performance of multimodal representation learning. Experimental evaluations on the MUStARD, CMU-MOSI, and CMU-MOSEI datasets demonstrate that our model consistently distills crucial information in multimodal learning scenarios, outperforming state-of-the-art benchmarks. Remarkably, on the CMU-MOSI dataset, ITHP surpasses human-level performance in the multimodal sentiment binary classification task across all evaluation metrics (i.e., Binary Accuracy, F1 Score, Mean Absolute Error, and Pearson Correlation).

## 1 Introduction

Perception represents the two-staged process of, first **(i)**, understanding, mining, identifying critical and essential cues, and translating multimodal noisy sensing (observations) into a world model. Second **(ii)**, defining the premises and foundations for causal reasoning, cognition, and decision-making in uncertain settings and environments. Consequently, a major focus in neuroscience refers to elucidating how the brain succeeds in quickly and efficiently mining and integrating information from diverse external sources and internal cognition to facilitate robust reasoning and decision-making (Tononi et al., 1998). Promising breakthroughs in understanding this complex process have emerged from exciting advancements in neuroscience research. Through a method of stepwise functional connectivity analysis (Sepulcre et al., 2012), the way the human brain processes multimodal data has been partially elucidated through recent research. The research outlined in Pascual-Leone & Hamilton (2001); Meunier et al. (2009); Zhang et al. (2019); Raut et al. (2020) suggests that information from different modalities forms connections in a specific order within the brain. This indicates that the brain processes multimodal information in a hierarchical manner, allowing for the differentiation between primary and secondary sources of information. Further research by (Mesulam, 1998; Lavenex & Amaral, 2000) elucidates that synaptic connections between different modality-receptive areas are reciprocal. This reciprocity enables information from various modalities to reciprocally exert feedback, thereby influencing early processing outcomes. Such a sequential

---

*Equal Contribution. †Corresponding to: `xiongyex@usc.edu` and `pbogdan@usc.edu`.

and hierarchical processing approach allows the brain to start by processing information in a certain modality, gradually linking and processing information in other modalities, and finally analyzing and integrating information effectively in a coordinated manner.

The brain selectively attends to relevant cues from each modality, extracts meaningful features, and combines them to form a more comprehensive and coherent representation of the multimodal stimulus. To exemplify this concept, let us consider sarcasm, which is often expressed through multiple cues simultaneously, such as a straight-looking face, a change in tone, or an emphasis on a word (Castro et al., 2019). In the context of human sarcasm detection, researchers have observed significant variation in sensitivity towards different modalities among individuals. Some are more attuned to auditory cues such as tone of voice, while others are more responsive to visual cues like facial expressions. This sensitivity establishes a primary source of information unique to each individual, creating a personalized perception of sarcasm. Simultaneously, other modalities aren't disregarded; instead, they reciprocally contribute, serving as supportive factors in the judgment process. In fact, researchers employ various experimental techniques, such as behavioral studies, neuroimaging (e.g., fMRI, Glover, 2011; PET, Price, 2012), and neurophysiological recordings, to investigate the temporal dynamics and neural mechanisms underlying multimodal processing in the human brain. The insights gained from these areas of neuroscience have been instrumental in guiding our novel approach to information integration within the scope of multimodal learning.

In multimodal learning, integrating different information sources is crucial for a thorough understanding of the surrounding phenomena or the environment. Although highly beneficial for a wide variety of autonomous systems and cyber-physical systems (CPS) Xue & Bogdan (2017), effectively understanding the complex interrelations among modalities is difficult due to their high-dimensionality and intricate correlations(Yin et al., 2023). To tackle this challenge, advanced modeling techniques are needed to fuse information from different modalities, enabling the discovery and utilization of rich interactions among them. Existing works have employed various approaches to fuse information from different modalities, treating each modality separately and combining them in different ways (Castro et al., 2019; Rahman et al., 2020). However, it has been seen that these fusion approaches deviate from the information-processing mechanism of neural synapses in the human brain. The above neuroscience research inspired us to devise a novel fusion method that constructs a hierarchical information perception model. Unlike conventional approaches that directly combine various modal information simultaneously, our method treats the prime modality as the input. We introduce a novel biomimetic model, referred to as Information-Theoretic Hierarchical Perception (ITHP), for effectively integrating and compacting multimodal information. Our objective is to distill the most valuable information from multimodal data, while also minimizing data dimensions and redundant information. Drawing inspiration from the information bottleneck (IB) principle proposed by Tishby et al. (Tishby et al., 2000), we propose a novel strategy involving the design of hierarchical latent states. This latent state serves to compress one modal state while retaining as much relevant information as possible about other modal states. By adopting the IB principle, we aim to develop compact representations that encapsulate the essential characteristics of multimodal data.

## 1.1 OUR CONTRIBUTIONS

In this paper, we propose the novel Information-Theoretic Hierarchical Perception (ITHP) model, which employs the IB principle to construct compact and informative latent states (information flow) for downstream tasks. The primary contributions of this research can be summarized as follows:

- We provide a novel insight into the processing of multimodal data by proposing a mechanism that designates the prime modality as the sole input, while linking it to other modal information using the IB principle. This approach offers a unique perspective on multimodal data fusion, reflecting a more neurologically-inspired model of information processing.

- We design ITHP model on top of recent neural network architectures, enhancing its accessibility and compatibility with most existing multimodal learning solutions.

- Remarkably, our ITHP-DeBERTa[1] framework outperforms human-level benchmarks in multimodal sentiment analysis tasks across multiple evaluation metrics. The model yields an 88.7% binary accuracy, 88.6% F1 score, 0.643 MAE, and 0.852 Pearson correlation,

---

[1]DeBERTa is an advanced variant of the BERT model for natural language embedding.

outstripping the human-level benchmarks of 85.7% binary accuracy, 87.5% F1 score, 0.710 MAE, and 0.820 Pearson correlation.

## 1.2 BACKGROUND AND RELATED WORK

**Information bottleneck.** Information bottleneck (IB) has emerged as a significant concept in the field of information theory and machine learning, providing a novel framework for understanding the trade-off between compression and prediction in an information flow. The IB principle formulates the problem of finding a compact representation of one given state while preserving its predictive power with respect to another state. This powerful approach has found numerous applications in areas such as speech recognition (Hecht & Tishby, 2005), document classification (Slonim & Tishby, 2000), and gene expression (Friedman et al., 2001). Moreover, researchers have successfully applied the IB principle to deep learning architectures (Shwartz-Ziv & Tishby, 2017; Tishby & Zaslavsky, 2015), shedding light on the role of information flow and generalization in these networks. In this paper, we employ the principles of IB to construct an effective information flow by designing a hierarchical perception architecture, paving the way for enhanced performance in downstream tasks.

**Multimodal learning.** Multimodal learning has been particularly instrumental in understanding language across various modalities - text, vision, and acoustic - and has been applied extensively to tasks such as sentiment analysis (Poria et al., 2018), emotion recognition (Zadeh et al., 2018d), personality traits recognition (Park et al., 2014), and chart understanding (Zhou et al., 2023). Work in this domain has explored a range of techniques for processing and fusing multimodal data. These include tensor fusion methods (Liu et al., 2018; Zadeh et al., 2017; 2016), attention-based cross-modal interaction methods (Lv et al., 2021; Tsai et al., 2019), and multimodal neural architectures (Pham et al., 2019; Hazarika et al., 2018), among others. Such explorations have given rise to advanced models like Tensor Fusion Networks (Zadeh et al., 2018a), Multi-attention Recurrent Network (Zadeh et al., 2018b), Memory Fusion Networks (Zadeh et al., 2018a), Recurrent Memory Fusion Network (Liang et al., 2018), and Multimodal Transformers (Tsai et al., 2019). These models utilize hierarchical fusion mechanisms, attention mechanisms, and separate modality-specific Transformers to selectively process information from multiple input channels. Despite significant strides in this domain, challenges persist, particularly when dealing with unaligned or limited and noisy multimodal data occurring in multi-agent systems navigating in uncertain unstructed environments. To surmount these obstacles, researchers have started exploring data augmentation methods and information-theoretic approaches to enrich multimodal representations and boost overall system performance.

## 2 MULTIMODAL LEARNING BASED ON INFORMATION BOTTLENECK

In this section, we introduce a novel method, called Information-Theoretic Hierarchical Perception (ITHP), for integrating and compressing multimodal noisy data streams. Taking inspiration from neuroscience research, our approach models the fusion and distillation of the information from multimodal information sources as a hierarchical structure, with a designated prime modality serving as the input. To link the input information with information from other modalities, we apply

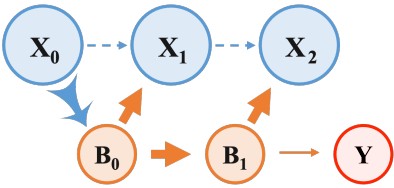

Figure 1: Constructing two latent states, $B_0$ and $B_1$, facilitates the transfer of pertinent information among three modal states $X_0$, $X_1$, and $X_2$.

the principle of Information Bottleneck (IB), ensuring an optimized compact representation of relevant information between modalities. Here, we begin by presenting a concise overview of the IB principle, followed by the construction of the hierarchical structure for multimodal information fusion. A typical example in multimodal information integration problems is the 3-modalities problem. We use this case as a practical illustration of the efficacy of our proposed model. To provide a detailed explanation, we initially formulate an optimization problem that delineates the specific question. Following this, we leverage the concept of the information bottleneck to build the structure of the ITHP model, which is then trained through the application of neural networks. Importantly, the ITHP model possesses the flexibility to be expanded to handle a multimodal context encompassing $N$

modalities. The exact model for this extension, along with a step-by-step derivation process, can be found in Appendix C for reference.

**Information Bottleneck (IB) approach.** The IB approach compresses a state $X$ into a new latent state $B$, while maintaining as much relevant information as possible about another state of interest $Y$. The latent state $B$ operates to minimize the remaining information of the compression task and to maximize the relevant information. Provided that the relevance of one state to another is often measurable by an information-theoretic metric (i.e., mutual information), the trade-off can be written as the following variational problem:

$$\min_{p(B|X)} I\left(X;B\right) - \beta I\left(B;Y\right). \tag{1}$$

The trade-off between the tasks mentioned above is controlled by the parameter $\beta$. In our model of ITHP, the latent states, solving the minimization problem (1), encodes the most informative part from the input states about the output states. Building upon this concept, we extend this one level of IB structure to create a hierarchy of bottlenecks, enabling the compact and accurate fusion of the most relevant information from multimodal data.

## 2.1 PROBLEM FORMULATION

From an intuitive perspective, fusing the multimodal information in a compact form is an optimization between separating the irrelevant or redundant information and preserving/extracting the most relevant information. The multimodal information integration problem for 3 modalities is shown in Fig. 1. In the information flow, we can think that $X_0$, $X_1$ and $X_2$ are three random variables that are not independent. We assume that the redundancy or richness of the information contained in the three modal states $X_0$, $X_1$, and $X_2$ is decreasing. Therefore, the system can start with the information of $X_0$ and gradually integrate the relevant information of $X_1$ and $X_2$ according to the hierarchical structure. In this configuration, we aim to determine the latent state $B_0$ that efficiently compresses the information of $X_0$ while retaining the relevant information of $X_1$. Moreover, $B_0$ acts as a pathway for conveying this information to $B_1$. Next, $B_1$ quantifies the meaningful information derived from $B_0$, representing the second-level latent state that retains the relevant information of $X_2$. Instead of directly fusing information based on the original $d_{X_0}$-dimensional data, we get the final latent state with a dimension of $d_{B_1}$ as the integration of the most relevant parts of multimodal information, which is then used for accurate prediction of $Y$.

The trade-off between compactly representing the primal modal states and preserving relevant information is captured by minimizing the mutual information while adhering to specific information processing constraints. Formally, we formulate this as an optimization problem:

$$\min_{p(B_0|X_0),p(B_1|B_0)} I(X_0;B_0) \qquad \text{subject to} \qquad \begin{aligned} I(X_0;X_1) - I(B_0;X_1) &\leq \epsilon_1 \\ I(B_0;B_1) &\leq \epsilon_2 \\ I(X_0;X_2) - I(B_1;X_2) &\leq \epsilon_3, \end{aligned} \tag{2}$$

where $\epsilon_1, \epsilon_2, \epsilon_3 > 0$. Our goal is to create hidden states, like $B_0$ and $B_1$, that condense the data from the primary modal state while preserving as much pertinent information from other modal states as possible. Specifically, we hope that the constructed hidden state $B_0$ covers as much relevant information in $X_1$ as possible, so $I(X_0;X_1) - \epsilon_1$ is a lower bound of $I(B_0;X_1)$. Similarly, $B_1$ is constructed to retains as much relevant information in $X_2$ as possible, with $I(X_0;X_2) - \epsilon_3$ as a lower bound. Additionally, we want to minimize $I(B_0;B_1)$ to ensure that we can further compress the information of $B_0$ into $B_1$.

## 2.2 INFORMATION-THEORETIC HIERARCHICAL PERCEPTION (ITHP)

This section takes the 3-modalities information integration problem as an example to illustrate ITHP as shown in Fig. 2. It introduces a hierarchical structure of IB levels combined with a multimodal representation learning task. Inspired by the deep variational IB approach proposed in Lee & van der Schaar (2021), we formulate the loss function incorporating the IB principle.

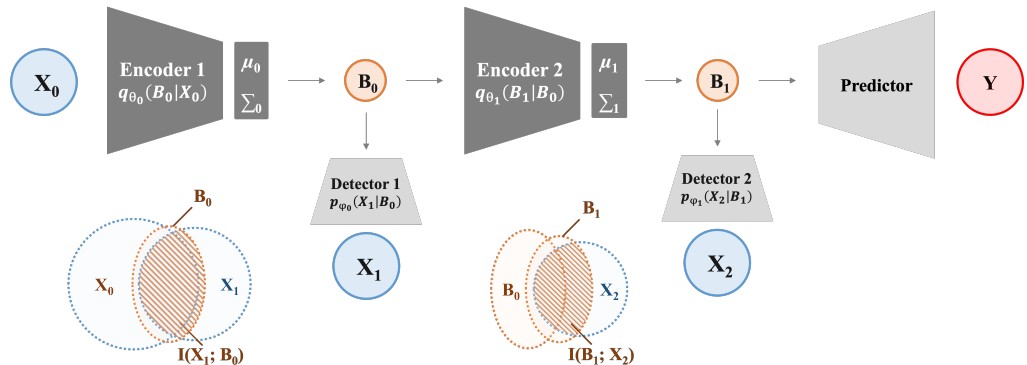

Figure 2: **An illustration of the proposed model architecture.** In each encoder, we have two MLP layers: the initial layer extracts the feature vectors from input states, while the second layer generates parameters for the latent Gaussian distribution. The Venn diagrams illustrate the information constraint from the optimization problem (2).

To address the optimization problem (2), we construct a Lagrangian function, incorporating the balancing parameters $\beta$, $\lambda$, and $\gamma$ to impose information processing constraints. Consequently, we solve the problem by minimizing the following function, which serves as the overall system loss function for training the network:

$$\mathcal{F}[p(B_0|X_0), p(B_0), p(X_1|B_0), p(X_2|B_0), p(B_1|B_0), p(B_1)] =$$
$$I(X_0; B_0) - \beta I(B_0; X_1) + \lambda\left(I(B_0; B_1) - \gamma I(B_1; X_2)\right). \tag{3}$$

We derive the loss function in Eqn. (3) based on the IB principle expressed in Eqn. (1) as a two-level structure of the Information Bottleneck. By minimizing the mutual information $I(X_0; B_0)$ and $I(B_0; B_1)$, and maximizing the mutual information $I(B_0; X_1)$ and $I(B_1; X_2)$, based on the optimization setup in Eqn. (2) and given balancing parameters, we ensure that this loss function is minimized during the neural network training process.

The ITHP model dealing with 3-modalities problems contains two levels of information bottleneck structures. We define $B_0$ as a latent state that compactly represents the information of the input $X_0$ while capturing the relevant information contained in $X_1$. The latent state $B_0$ is obtained through a stochastic encoding with a Neural Network, denoted as $q_{\theta_0}(B_0|X_0)$, aiming to minimize the mutual information between $X_0$ and $B_0$. Given the latent state representation $B_0$ obtained from the encoding Neural Network, the outputting MLP plays the role of capturing the relevant information of $X_1$ through $q_{\psi_0}(X_1|B_0)$. Similarly, we have the latent state $B_1$ as a stochastic encoding of $B_0$, achieved through $q_{\theta_1}(B_1|B_0)$, and an output predictor to $X_2$ through $q_{\psi_1}(X_2|B_1)$. Throughout the rest of the paper, we denote the true and deterministic distribution as $p$ and the random probability distribution fitted by the neural network as $q$. Here, $\theta_0$ and $\theta_1$ denote the parameters of encoding neural networks while $\psi_0$ and $\psi_1$ denote the parameters of the output predicting neural networks. The following loss functions are defined based on the IB principle:

$$\mathcal{L}_{IB_0}^{\theta_0,\psi_0}(X_0; X_1) = I(X_0; B_0) - \beta I(B_0; X_1)$$
$$\approx \mathbb{E}_{X_0 \sim p(X_0)} KL\left(q_{\theta_0}(B_0|X_0)\,||\,q(B_0)\right) \tag{4}$$
$$- \beta \cdot \mathbb{E}_{B_0 \sim p(B_0|X_0)} \mathbb{E}_{X_0 \sim p(X_0)}\left[\log q_{\psi_0}(X_1|B_0)\right],$$

$$\mathcal{L}_{IB_1}^{\theta_1,\psi_1}(B_0; X_2) = I(B_0; B_1) - \gamma I(B_1; X_2)$$
$$\approx \mathbb{E}_{B_0 \sim p(B_0)} KL\left(q_{\theta_1}(B_1|B_0)\,||\,q(B_1)\right) \tag{5}$$
$$- \gamma \cdot \mathbb{E}_{B_1 \sim p(B_1|B_0)} \mathbb{E}_{B_0 \sim p(B_0)}\left[\log q_{\psi_1}(X_2|B_1)\right],$$

where $\beta, \gamma \geq 0$ are balancing coefficients chosen to reflect the trade-off between two information quantities. Here, $KL\left(p(X)\,||\,q(X)\right)$ is the Kullback-Leibler (KL) divergence between the two distributions $p(X)$ and $q(X)$. Detailed derivations of the Eqns. (4) and (5) are presented in Appendix C.1, while the extension from the 3-modalities problem to $N$-modalities has been elaborated in Appendix C.2.

## 2.3 Training method of ITHP

We train the consecutive two-level hierarchy ITHP with an overall loss function labeled with $X_0$ and $X_2$. To minimize the overall loss, we introduce the hyper-parameter $\lambda > 0$ that balances the trade-off between the losses of the two-level hierarchy:

$$\mathcal{L}_{overall}^{\theta,\psi}(X_0, X_2) = \mathcal{L}_{IB_0}^{\theta_0,\psi_0}(X_0; X_1) + \lambda \cdot \mathcal{L}_{IB_1}^{\theta_1,\psi_1}(B_0; X_2). \qquad (6)$$

In the context of multimodal learning, we apply the final fused latent state $B_{k+1}$ to the processing of representation learning tasks. Specifically, the final loss function combines the two-level hierarchy loss with a task-related loss function:

$$\mathcal{L} = \frac{2}{\beta + \gamma} \cdot \mathcal{L}_{overall}^{\theta,\psi}(X_0, X_2) + \alpha \cdot \mathcal{L}_{task-related}(B_1, Y). \qquad (7)$$

The Eqn. (7) introduces a new parameter $\alpha \geq 0$ to weigh the loss function of multimodal information integration and the loss function of processing downstream tasks. The pseudo-code of the ITHP algorithm is provided in Appendix D.

## 3 Experiments

In this section, we evaluate our proposed Information-Theoretic Hierarchical Perception (ITHP) model on three popular multimodal datasets: the Multimodal Sarcasm Detection Dataset (**MUStARD**; Castro et al., 2019), the Multimodal Opinion-level Sentiment Intensity dataset (**MOSI**; Zadeh et al., 2016), and the Multimodal Opinion Sentiment and Emotion Intensity (**CMU-MOSEI**; Zadeh et al., 2018d). Detailed descriptions of the datasets can be found in Appendix F. We demonstrate that the ITHP model is capable of constructing efficient and compact information flows. In the sentiment analysis task, we show that the ITHP-DeBERTa framework outperforms human-level benchmarks across multiple evaluation metrics, including binary accuracy, F1 score, MAE, and Pearson correlation. All experiments were conducted on Nvidia $A100$ 40GB GPUs. Detailed information regarding the model architecture and training parameters used in each experiment can be found in Appendix E.

### 3.1 Sarcasm detection

In the sarcasm detection task, to ensure a fair comparison, we utilize the same embedding data as provided in Castro et al. (2019). The embedding data was produced through multiple methods. Textual utterance features were derived from sentence-level embeddings using a pre-trained BERT model (Devlin et al., 2018), which provides a representation of size $d_t = 768$. For audio features, the Librosa library (McFee et al., 2018a) was utilized to extract low-level features such as MFCCs, Mel spectrogram, spectral centroid, and their deltas (McFee et al., 2018b), which are concatenated together to compose a $d_a = 283$ dimensional joint representation. For video features, the frames in the utterance are processed using a `pool5` layer of an ImageNet (Deng et al., 2009) pre-trained ResNet-152 model (He et al., 2016), which provides a representation of size $d_v = 2048$.

In addition to providing the multimodal embedding data, Castro et al. (2019) also conducted experiments using their own fine-tuned multimodal sarcasm detection model (referred to as "MSDM" in this paper). These experiments demonstrated that the integration of multimodal information can enhance the automatic classification of sarcasm. The results of their sarcasm detection task, as reported in Table 1 (MSDM), serve as a benchmark for comparing the performance of our proposed model. The classification results for sarcasm detection, obtained with different combinations of modalities (Text - $T$, Audio - $A$, Video - $V$), provide a basis for the comparative analysis conducted in our study.

**Varying combinations of modalities:** To test for the unimodal data, we initially established a two-layer neural network to serve as a predictor for the downstream task. The binary classification results are evaluated by weighted precision, recall, and F-score across both sarcastic and non-sarcastic classes. The evaluation is performed using a 5-fold cross-validation approach to ensure robustness and reliability. When evaluating each individual modality using the predictor, the results are comparable to those of the MSDM, with metrics hovering around 65%. This finding indicates that the embedding data from each modality contains relevant information for sarcasm detection. However, depending solely on a single modality appears to restrict the capacity for more accurate predictions. Considering the binary classification results across the three modalities and taking into account the size of the

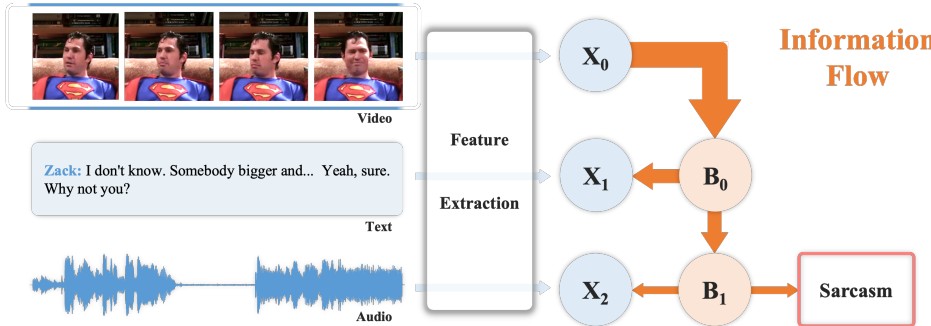

Figure 3: **A schematic representation of our proposed ITHP and its information flow.** The diagram illustrates the process of feature extraction from multimodal embedding data including video frames, text, and audio patterns. These modalities pass through a "Feature Extraction" phase, where they are embedded to get modal states $X_0$, $X_1$, and $X_2$. The derived states are then processed to construct latent states $B_0$ and $B_1$. This processing includes reciprocal information exchange between $X_1$ and $B_0$, as well as between $B_1$ and $X_2$. The resulting information from this process is then used to make a determination about the presence of sarcasm.

| Algorithm | Metrics | Modalities | | | | | | |
|---|---|---|---|---|---|---|---|---|
| | | **T** | **A** | **V** | **T-A** | **V-T** | **V-A** | **V-T-A** |
| MSDM | Precision | 65.1 | 65.9 | 68.1 | 66.6 | 72.0 | 66.2 | 71.9 |
| | Recall | 64.6 | 64.6 | 67.4 | 66.2 | 71.6 | 65.7 | 71.4 |
| | F-Score | 64.6 | 64.6 | 67.4 | 66.2 | 71.6 | 65.7 | 71.5 |
| ITHP | Precision | 65.0 | 65.1 | 67.6 | 69.7 | 72.1 | 70.5 | **75.3** |
| | Recall | 64.8 | 64.9 | 67.5 | 69.1 | 71.7 | 70.3 | **75.2** |
| | F-Score | 64.7 | 64.8 | 67.5 | 69.1 | 71.6 | 70.3 | **75.2** |

Table 1: **Results of sarcasm detection on the MUStARD dataset.** The table presents the weighted Precision, Recall, and F-Score across both sarcastic and non-sarcastic classes, averaged across five folds. Underlined values indicate the best results for each row, while bold data represents the optimal results overall. The modalities are denoted as follows: $T$ represents text, $A$ stands for audio, and $V$ signifies video.

embedding features for each modality ($d_v = 2048, d_t = 768, d_a = 283$), we designate $V$ as the prime modality and construct the consecutive modality states of $V(X_0) - T(X_1) - A(X_2)$. During the evaluation of two-modal combinations, MSDM utilizes concatenated data to represent the combination, whereas we employ the single-stage information theoretical perception (ITP) approach, which involves only one latent state, denoted as $B_0$.

For the two-modal learning, there are three different combinations $T(X_0) - A(X_1)$, $V(X_0) - T(X_1)$, and $V(X_0) - A(X_1)$. In the case of $T(X_0) - A(X_1)$ and $V(X_0) - T(X_1)$, both MSDM and ITHP improved performance compared to using a single modality, indicating the advantages of leveraging multimodal information. However, it is worth noting that the result of the combination $V(X_0) - A(X_1)$ is even lower than the result on $V(X_0)$ from MSDM, suggesting that the model struggled to effectively extract meaningful information by combining the embedding features from video and audio. In contrast, ITP successfully extracts valuable information from these two modalities, resulting in higher metric values. For the combination $V(X_0) - T(X_1) - A(X_2)$, when considering the metrics of weighted precision, recall, and F-score, compared to the best-performing unimodal learning, MSDM's improvements are 5.58%, 5.93%, and 6.08%, respectively; whereas our ITHP model shows the improvements of 11.39%, 11.41% and 11.41% respectively. These results indicate that our ITHP model succeeds to construct the effective information flow among the multimodality states for the sarcasm detection task. Compared to the best-performing 2-modal learning, MSDM shows lower performance in the 3-modal learning setting, while our ITHP model demonstrates improvements of 4.44% in precision, 4.88% in the recall, and 5.03% in F-score, highlighting the advantages of the hierarchical architecture employed in our ITHP model as shown in Fig. 3.

**Varying Lagrange multiplier:** To conduct a comprehensive analysis, we explore the mutual information among these modalities. In the IB method, the Lagrange multiplier controls the trade-off between two objectives: (i) maximizing the mutual information between the compressed representation $B$ and the relevant variable $Y$ to ensure that the pertinent information about $Y$ is retained in $B$, and (ii)

minimizing the mutual information between the compressed representation $B$ and the original data $X$ to eliminate redundant details in $X$.

A high value of the Lagrange multiplier encourages the model to retain more relevant information about $Y$. In our ITHP model learning for 3 modalities, we have two Lagrange multipliers $\beta$ and $\gamma$ to regulate the balance between the mutual information of the latent states and multimodal states. Here, we adjust the values of the Lagrange multipliers to observe their influence on the overall performance of the ITHP model for the sarcasm detection task, and the results are shown in Fig. 4. When $\beta = 2$ and $\gamma = 2$, the model shows the worst performance with a weighted precision of $0.619$ and a recall of $0.607$, which indicates that the model cannot retain the relevant information from the other modalities while losing the information from the input state with a low Lagrange multiplier, leading to even worse performance than the unimodal learning.

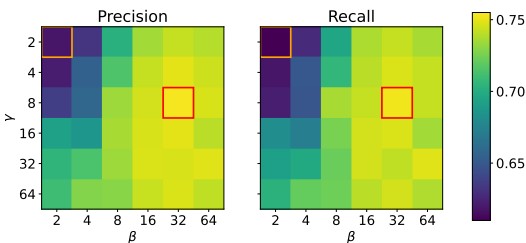

Figure 4: **Weighted precision and recall for the binary classification task under varying Lagrange multipliers.** The graph shows the impact of varying the Lagrange multipliers ($\beta$ and $\gamma$). For each plot, the Red (Orange) color denotes the highest (lowest) value, respectively.

The best performance with a precision of $0.753$ and recall of $0.752$ comes when $\beta = 32$ and $\gamma = 8$. The region around the point is also the brightest $3 \times 3$ square with all values above $0.74$, which indicates an optimal balance to construct the most effective information flow among the varying values of $\beta$ and $\gamma$. It is worth noting that the upper right triangular part is brighter than the lower left triangular part, which shows that the model benefits more from a higher $\beta$ than a higher $\gamma$. It suggests that retaining more relevant information from Text ($X_1$) is more important for sarcasm detection, which is also verified by the best performance of the combination $V(X_0) - T(X_1)$ among all the two modality combinations.

## 3.2 SENTIMENT ANALYSIS

Sentiment Analysis (SA) has undergone substantial advancements with the introduction of multimodal datasets, incorporating text, images, and videos Gandhi et al. (2022). Our study concentrates on two pivotal datasets: the Multimodal Corpus of Sentiment Intensity (MOSI) and Multimodal Opinion Sentiment and Emotion Intensity (MOSEI). Notably, the MOSI dataset employs a unique feature extraction process compared to sarcasm detection, with embedding feature sizes of $d_t = 768$, $d_a = 74$, and $d_v = 47$ for text, audio, and video, respectively. We hypothesize that the text embedding features $T$ hold the most substantial information, followed by $A$, leading to the formulation of consecutive modality states of $T(X_0) - A(X_1) - V(X_2)$ for this task.

In evaluating the versatility of our approach across diverse multimodal language datasets, experiments are also conducted on the CMU-MOSEI dataset (Zadeh et al., 2018d). The

| | CMU-MOSI | | | |
|---|---|---|---|---|
| **Task Metric** | **BA↑** | **F1↑** | **MAE↓** | **Corr↑** |
| | BERT | | | |
| Self-MM$_b$ | 84.0 | 84.4 | 0.713 | 0.798 |
| MMIM$_b$ | 84.1 | 84.0 | 0.700 | 0.800 |
| MAG$_b$ | 84.2 | 84.1 | 0.712 | 0.796 |
| Self-MM$_d$ | 55.1 | 53.5 | 1.44 | 0.158 |
| MMIM$_d$ | 85.8 | 85.9 | 0.649 | 0.829 |
| MAG$_d$ | 86.1 | 86.0 | 0.690 | 0.831 |
| UniMSE | 85.9 | 85.8 | 0.691 | 0.809 |
| MIB | 85.3 | 85.3 | 0.711 | 0.798 |
| BBFN | 84.3 | 84.3 | 0.776 | 0.755 |
| **ITHP** | **88.7** | **88.6** | **0.643** | **0.852** |
| Human | 85.7 | 87.5 | 0.710 | 0.820 |

Table 2: **Model performance on CMU-MOSI dataset.** The table presents the performance of other BERT-based (indicated with subscript "$b$") and DeBERTa-based (indicated with subscript "$d$") pre-trained models. Models developed by us are highlighted in bold, and optimal results are underlined.

CMU-MOSEI dataset, while resembling MOSI in annotating utterances with sentiment intensity, places a greater emphasis on positive sentiments. Additionally, it provides annotations for nine discrete and three continuous emotions, expanding its applicability for various affective computing tasks Das & Singh (2023); Liang et al. (2023).

Our analysis encompasses a range of prominent models, including Self-MM (Yu et al., 2021), MMIM (Han et al., 2021b), MAG (Rahman et al., 2020), MIB (Mai et al., 2022), BBFN (Han et al., 2021a), and UniMSE (Hu et al., 2022). Please refer to Appendix B for details.

Delving into framework integration, this research incorporates ITHP with pre-trained DeBERTa models, optimizing textual data processed by DeBERTa's Embedding and Encoder layers, unlike the MAG model which utilizes embedding data from BERT's middle layer. A significant development introduced is ITHP-DeBERTa, integrating the Decoding-enhanced BERT with Disentangled Attention (DeBERTa; He et al., 2020; 2021). This advanced model enhances BERT and RoBERTa (Liu et al., 2019) through disentangled attention and a sophisticated mask decoder.

The results on the CMU-MOSI dataset are reported in Table 2. Additionally, Table 4 in Appendix F.1 provides reference results for more DeBERTa-based baseline models. As shown in the results, our ITHP model outperforms all the SOTA models in both BERT and DeBERTa incorporation settings.

It is noteworthy that significant strides in performance were observed with the integration of DeBERTa with the ITHP model, surpassing even human levels. This integration with DeBERTa resulted in a $3.5\%$ increase in Binary Classification Accuracy (BA), $1.3\%$ improvement in F1 Score (F1), a reduction of $9.4\%$ in Mean Absolute Error (MAE), and a $3.9\%$ increase in Pearson Correlation (Corr). To explore the applicability of our approach to various datasets, we perform experiments on the extensive CMU-MOSEI dataset. The results in Table 3 demenstrate that our ITHP model consistently surpasses the SOTA, thereby confirming its robustness and generalizability. In contrast, it is worth noting that Self-MM itself heavily relies on the feature extraction process performed by BERT, resulting in a significant degradation of DeBERTa-based Self-MM on both CMU-MOSI and CMU-MOSEI datasets, which restricts its applicability in certain scenarios.

| CMU-MOSEI | | | | |
|---|---|---|---|---|
| **Task Metric** | **BA↑** | **F1↑** | **MAE↓** | **Corr↑** |
| Self-MM$_b$ | 85.0 | 85.0 | 0.529 | 0.767 |
| MMIM$_b$ | 86.0 | 86.0 | 0.526 | 0.772 |
| MAG$_b$ | 84.8 | 84.7 | 0.543 | 0.755 |
| Self-MM$_d$ | 65.3 | 65.4 | 0.813 | 0.208 |
| MMIM$_d$ | 85.2 | 85.4 | 0.568 | 0.799 |
| MAG$_d$ | 85.8 | 85.9 | 0.636 | 0.800 |
| **ITHP** | **87.3** | **87.4** | **0.564** | **0.813** |

Table 3: **Model performance on CMU-MOSEI dataset.** Models developed by us are highlighted in bold, and optimal results are underlined.

## 4 CONCLUSION

Drawing inspiration from neurological models of information processing, we build the links between different modalities using the information bottleneck (IB) method. By leveraging the IB principle, our proposed ITHP constructs compact and informative latent states for information flow. The hierarchical architecture of ITHP enables incremental distillation of useful information with applications in perception of autonomous systems and cyber-physical systems. The empirical results emphasize the potential value of the advanced pre-trained large language models and the innovative data fusion techniques in multimodal learning tasks. When combined with DeBERTa, our proposed ITHP model is the first work, based on our knowledge, to outperform human-level benchmarks on all evaluation metrics (i.e., BA, F1, MAE, Corr).

## 5 LIMITATIONS AND FUTURE WORK

A potential challenge of our model is its reliance on a preset order of modalities. We've illustrated this aspect with experiments showcasing various modalities' orders in Appendix Table 11 in Appendix F.5. Typically, prior knowledge is utilized to select the primary modality and rank others. Without any knowledge, addressing this challenge by adjusting the Lagrange multipliers or iterating through all possible modality orders can be very time-consuming. Another potential challenge emerges when the primary modality lacks sufficient information independently, necessitating complementary information from other modalities for optimized performance. Nonetheless, we believe that the integration of deep learning architecture into the hierarchical perception can mitigate this issue. For a more detailed discussion, please refer to Appendix H. Future work will focus on addressing the identified challenges by exploring how to utilize deep learning frameworks to achieve efficient neuro-inspired modality ordering.

ACKNOWLEDGEMENT

The authors acknowledge the support by the U.S. Army Research Office (ARO) under Grant No. W911NF-23-1-0111, the National Science Foundation (NSF) under the Career Award CPS-1453860, CCF-1837131, MCB-1936775, CNS-1932620 and the NSF award No. 2243104 under the Center for Complex Particle Systems (COMPASS), the Defense Advanced Research Projects Agency (DARPA) Young Faculty Award and DARPA Director Award under Grant Number N66001-17-1-4044, an Intel faculty award and a Northrop Grumman grant. The authors are deeply grateful to Nian Liu, Ruicheng Yao, and Xin Ren from Tuyou Travel AI for their efforts in promoting industry integration and providing both computing support and helpful feedback. The views, opinions, and/or findings in this article are those of the authors and should not be interpreted as official views or policies of the Department of Defense or the National Science Foundation.

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

## A    CODE AVAILABILITY

Our codebase can be found in [https://github.com/joshuaxiao98/ITHP]. Our code uses assets from MAG-BERT (Rahman et al., 2020), BERT (Devlin et al., 2018) [https://github.com/google-research/bert, Apache License 2.0], and DeBERTa (He et al., 2020; 2021) [https://github.com/microsoft/DeBERTa, MIT License].

## B    RELATED WORK

Multimodal sentiment analysis has been a prominent research area, focusing on the integration of verbal and nonverbal information across different modalities such as text, acoustics, and vision. Prior studies in this field have primarily concentrated on representation learning and multimodal fusion approaches. For representation learning, Wang et al. (Wang et al., 2019) proposed a recurrent attended variation embedding network to learn the shifted word representations, while Hazarika et al. (Hazarika et al., 2020) introduced modality-invariant and modality-specific representations. In terms of multimodal fusion, models can be categorized into early fusion and late fusion methods. Early fusion models incorporate delicate attention mechanisms for cross-modal fusion, as demonstrated by Zadeh et al. (Zadeh et al., 2018a) who developed a memory fusion network for cross-view interactions, and Tsai et al. (Tsai et al., 2019) who proposed cross-modal transformers to enhance a target modality through cross-modal attention. Late fusion methods, on the other hand, prioritize intra-modal representation learning before inter-modal fusion. For instance, Zadeh et al. (Zadeh et al., 2017) utilized a tensor fusion network to compute the outer product between unimodal representations, while Liu et al. (Liu et al., 2018) introduced a low-rank multimodal fusion method to reduce computational complexity.

In recent years, the fusion methods, combined with advanced Transformer-based large language models such as BERT (Devlin et al., 2018) and XLNet (Yang et al., 2019), have achieved remarkable progress in multimodal sentiment analysis. These methods have reached near-human level performance on the CMU-MOSI dataset in the task of multimodal sentiment binary classification (Rahman et al., 2020). Innovations such as Self-MM (Yu et al., 2021) have introduced self-supervised strategies that refine unimodal label learning, while MMIM (Han et al., 2021b) utilizes mutual information to enhance sentiment discernment from multimodal data. The MAG approach (Rahman et al., 2020) integrates affective gaps between modalities, and MIB (Mai et al., 2022) focuses on isolating essential data features, filtering out the extraneous for purer multimodal fusion. BBFN (Han et al., 2021a) reimagines the fusion process by merging and differentiating between modal inputs. The current SOTA model presents novel insights, UniMSE (Hu et al., 2022) innovatively integrates multimodal sentiment analysis and emotion recognition in conversations into a unified framework, enhancing prediction accuracy by leveraging the synergies and complementary aspects of sentiments and emotions. The SPECTRA(Yu et al., 2023) is pioneering in spoken dialog understanding by pre-training on speech and text, featuring a unique temporal position prediction task for precise speech-text alignment.The DynMM (Xue & Marculescu, 2023) creates the possibility of reducing computing resource consumption. By introducing dynamic multi-modal fusion, which fuses input data from multiple modalities adaptively, leading to reduced computation, improved representation power, and enhanced robustness.

Additionally, the latest work in the field of sarcasm detection has expanded our research horizons. The HFM (Cai et al., 2019a) utilizes early fusion and representation fusion techniques to refine feature representations for each modality. It integrates information from various modalities to better utilize the available data and enhance sarcasm detection performance. The DIP(Wen et al., 2023) creatively adopts dual incongruity perception at factual and affective levels, enabling effective detection of sarcasm in multi-modal data. Aside from multimodal sentiment analysis, numerous applications involve the integration of multiple modal inputs (Tian et al., 2022; 2024; Huang et al., 2023; 2024; Xiao et al., 2022), making the management of inter-modal interactions a significant area of ongoing research interest.

## C DERIVATION OF THE VARIATIONAL IB LOSS FOR ITHP

### C.1 3-MODALITIES SCENARIO

In the realm of multimodal learning, which usually involves three useful modalities, we outline a loss function for a 3-modalities setting in our study. This scenario incorporates three consecutive modalities, $X_0$, $X_1$, and $X_2$, these three modalities are ranked according to the richness of main information. In this section, we derive the formula of the two-level information bottleneck hierarchy in the three-modal problem in detail, as shown in Eqn. (4) and Eqn. (5).

Equation (4) represents the loss of the input-level Information Bottleneck as $(X_0 - B_0 - X_1)$ in the system, while the Eqn. (5) represents a general expression of the subsequent-level Information Bottleneck as $(B_0 - B_1 - X_2)$, as shown in Fig. (1). In this section, we first show the detailed derivation of the IB loss function (4) and then give the derivation of Eqn. (5) in a concise manner. Firstly, the loss function of the first input-level Information Bottleneck can be given as:

$$\mathcal{L}_{IB_0}^{\theta_0, \psi_0}(X_0; X_1) = I(X_0; B_0) - \beta I(B_0; X_1),  \tag{8}$$

where $\beta > 0$ is a trading-off parameter chosen to balance between the two mutual information quantities. Here, $I(X_0; B_0)$ and $I(B_0; X_1)$ can be derived using variational approximations, i.e., $q_{\theta_0}(B_0|X_0)$ and $q_{\psi_0}(X_1|B_0)$, as follows:

$$
\begin{aligned}
I(X_0; B_0) &= \int dX_0 dB_0 p(X_0, B_0) \cdot \log \frac{p(X_0, B_0)}{p(X_0) \cdot p(B_0)} \\
&= \int dX_0 dB_0 p(X_0, B_0) \cdot \log \frac{p(B_0|X_0)}{p(B_0)} \\
&= \int dX_0 dB_0 p(X_0, B_0) \cdot \log p(B_0|X_0) - \int dB_0 p(B_0) \cdot \log p(B_0) \\
&\leq \int dX_0 dB_0 p(X_0, B_0) \cdot \log q_{\theta_0}(B_0|X_0) - \int dB_0 p(B_0) \cdot \log q(B_0) \\
&\approx \int dX_0 dB_0 p(X_0) \cdot q_{\theta_0}(B_0|X_0) \cdot \log \frac{q_{\theta_0}(B_0|X_0)}{q(B_0)} \\
&= \mathbb{E}_{X_0 \sim p(X_0)}\left[KL\left(q_{\theta_0}(B_0|X_0)||q(B_0)\right)\right],
\end{aligned}  \tag{9}
$$

where $q_{\theta_0}(B_0|X_0)$ is used as the variational approximation for $p(B_0|X_0)$ and $q(B_0) \sim \mathcal{N}(0,1)$ is the probability distribution over the latent states based on the assumption.

$$
\begin{aligned}
I(B_0; X_1) &= \int dB_0 dX_1 p(X_1, B_0) \cdot \log \frac{p(X_1, B_0)}{p(X_1) \cdot p(B_0)} \\
&= \int dB_0 dX_1 p(X_1, B_0) \cdot \log \frac{p(X_1|B_0)}{p(X_1)} \\
&\geq \int dB_0 dX_1 p(X_1, B_0) \cdot \log \frac{q_{\psi_0}(X_1|B_0)}{p(X_1)} \\
&\approx \int dB_0 dX_1 p(X_1, B_0) \cdot \log q_{\psi_0}(X_1|B_0) - \int dX_1 p(X_1) \cdot \log q(X_1) \\
&= \int dB_0 dX_1 p(X_1, B_0) \cdot \log q_{\psi_0}(X_1|B_0) + H(X_1),
\end{aligned}  \tag{10}
$$

where $H(X_1)$ is the entropy of $X_1$, which is determined only by the distribution of $X_1$ itself, so this can be ignored in the loss function. Since we cannot derive the distribution of $p(B_0|X_1)$ directly, we have to introduce the modal $X_0$ with the relationship that $B_0$ is independent of $X_1$ given $X_0$:

$$p(X_1, B_0) = \int dX_0 p(X_0, X_1, B_0) = \int dX_0 p(X_0) \cdot p(X_1|X_0) \cdot p(B_0|X_0).  \tag{11}$$

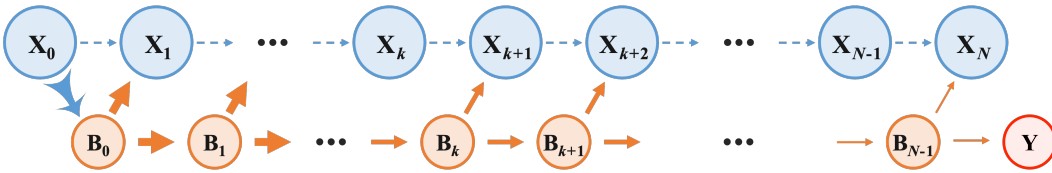

Figure 5: $N$ latent states are constructed for extracting and transferring the most relevant information of a set of $N + 1$ modal states. The modal states are represented by $X_0$, $X_1$, ..., $X_{N-1}$, and $X_N$. This order is determined by the richness of the amount of information contained in the modalities. The latent states $B_0$, $B_1$, ..., $B_{N-1}$ represent the pathway for transferring the relevant information among $X$. Through the hierarchical structure, the most relevant information from $X_0$ to $X_N$ is gradually retained and distilled.

Therefore, $I(B_0; X_1)$ can be given by

$$I(B_0; X_1) = \int dX_0 dB_0 dX_1 p(X_0) \cdot p(X_1|X_0) \cdot p(B_0|X_0) \cdot \log q_{\psi_0}(X_1|B_0)$$
$$= \mathbb{E}_{B_0 \sim p(B_0|X_0)} \mathbb{E}_{X_0 \sim p(X_0)} [\log q_{\psi_0}(X_1|B_0)], \tag{12}$$

where $q_{\psi_0}(B_0|X_0)$ are used as the variational approximation for $p(B_0|X_0)$. Then we have the derivation of the Eqn. (4) as

$$\mathcal{L}_{IB_0}^{\theta_0, \psi_0}(X_0; X_1) = I(X_0; B_0) - \beta I(B_0; X_1)$$
$$\approx \mathbb{E}_{X_0 \sim p(X_0)} KL(q_{\theta_0}(B_0|X_0) || q(B_0)) \tag{13}$$
$$- \beta \cdot \mathbb{E}_{B_0 \sim p(B_0|X_0)} \mathbb{E}_{X_0 \sim p(X_0)} [\log q_{\psi_0}(X_1|B_0)].$$

And we can easily get the derivation of the Eqn. (5) in similar steps as above. The loss function of the subsequent-level Information Bottleneck is shown as:

$$\mathcal{L}_{IB_1}^{\theta_1, \psi_1}(B_0; X_2) = I(B_0; B_1) - \gamma I(B_1; X_2). \tag{14}$$

The results of $I(B_0; B_1)$ and $I(B_1; X_2)$ are shown as:

$$I(B_0; B_1) = \int dB_0 dB_1 p(B_0, B_1) \cdot \log \frac{p(B_0, B_1)}{p(B_0) \cdot p(B_1)}$$
$$\approx \int dB_0 dB_1 p(B_0) \cdot q_{\theta_1}(B_1|B_0) \cdot \log \frac{q_{\theta_1}(B_1|B0)}{q(B_1)} \tag{15}$$
$$= \mathbb{E}_{B_0 \sim p(B_0)} [KL(q_{\theta_1}(B_1|B_0) || q(B_1))],$$

$$I(B_1; X_2) = \int dB_1 dX_2 p(X_2, B_1) \cdot \log \frac{p(X_2, B_1)}{p(X_2) \cdot p(B_1)}$$
$$\approx \int dB_1 dX_2 p(X_2, B_1) \cdot \log q_{\psi_1}(X_2|B_1) + H(X_2) \tag{16}$$
$$= \mathbb{E}_{B_1 \sim p(B_1|B_0)} \mathbb{E}_{B_0 \sim p(B_0)} [\log q_{\psi_1}(X_2|B_1)].$$

Then we have the derivation of the Eqn. (5) as

$$\mathcal{L}_{IB_1}^{\theta_1, \psi_1}(B_0; X_2) = I(B_0; B_1) - \gamma I(B_1; X_2)$$
$$\approx \mathbb{E}_{B_0 \sim p(B_0)} KL(q_{\theta_1}(B_1|B_0) || q(B_1)) \tag{17}$$
$$- \gamma \cdot \mathbb{E}_{B_1 \sim p(B_1|B_0)} \mathbb{E}_{B_0 \sim p(B_0)} [\log q_{\psi_1}(X_2|B_1)].$$

### C.2 GENERAL SCENARIO FOR N MODALITIES

In what follows, we present the loss function for scenarios involving more than 3 modalities, thus, a framework for a more general multimodal learning case. Let's consider a multi-modalities learning system with $N$ informative modalities, as shown in Fig. 5. According to the difference in information richness in $N$ modalities, we sort the information of these $N$ modalities, and ranked from high to low according to information richness as $X_0, X_1, \cdots\cdots, X_{N-1}, X_N$. Therefore, we start integrating the multimodal information from the modal state $X_0$ with the highest information richness. The input

level Information Bottleneck loss can be derived from Eqn. (4), which is the same as 3-modalities scenario. The loss function is shown as:

$$
\begin{aligned}
\mathcal{L}_{IB_0}^{\theta_0,\psi_0}\left(X_0;X_1\right) &= I\left(X_0;B_0\right) - \beta I\left(B_0;X_1\right) \\
&\approx \mathbb{E}_{X_0\sim p(X_0)} KL\left(q_{\theta_0}\left(B_0|X_0\right)||q\left(B_0\right)\right) \\
&- \beta \cdot \mathbb{E}_{B_0\sim p(B_0|X_0)}\mathbb{E}_{X_0\sim p(X_0)}\left[\log q_{\psi_0}\left(X_1|B_0\right)\right].
\end{aligned}
\tag{18}
$$

Then, the subsequent level IB loss can be given as:

$$
\begin{aligned}
\mathcal{L}_{IB_{k+1}}^{\theta_{k+1},\psi_{k+1}}\left(B_k;X_{k+2}\right) &= I\left(B_k;B_{k+1}\right) - \gamma_k I\left(B_{k+1};X_{k+2}\right) \\
&\approx \mathbb{E}_{B_k\sim p(B_k)} KL\left(q_{\theta_{k+1}}\left(B_{k+1}|B_k\right)||q\left(B_{k+1}\right)\right) \\
&- \gamma_k \cdot \mathbb{E}_{B_{k+1}\sim p(B_{k+1}|B_k)}\mathbb{E}_{B_k\sim p(B_k)}\left[\log q_{\psi_{k+1}}\left(X_{k+2}|B_{k+1}\right)\right], \\
&(k = 0,1,\cdots\cdots,N-2).
\end{aligned}
\tag{19}
$$

Then, we train the overall network using $N$ modal states. It is worth mentioning that in our model (ITHP), only $X_0$ with the highest information richness is used as the model input, and $X_1,\cdots\cdots,X_{N-1},X_N$ act as detectors in the information flow. To minimize the overall loss, we introduce a series of Lagrange multipliers $\lambda_k > 0$ that balance the trade-off between the losses of the input-level IB and all subsequent IB levels in the whole hierarchical structure. Therefore, the overall loss function mentioned in Eqn. (6) can be generalized to the form with $N$ modal states as:

$$
\mathcal{L}_{overall}^{\theta,\psi}\left(X_0,X_N\right) = \mathcal{L}_{IB_0}^{\theta_0,\psi_0}\left(X_0;X_1\right) + \sum_{k=0}^{N-2}\lambda_k \cdot \mathcal{L}_{IB_{k+1}}^{\theta_{k+1},\psi_{k+1}}\left(B_k;X_{k+2}\right).
\tag{20}
$$

With the overall loss derived here, we can further get the expression of the loss function considering the downstream task, so the Eqn. (7) can be generalized to the form with $N$ modal states as:

$$
\mathcal{L}_{task-related} = \frac{N-1}{\beta + \sum_{k=0}^{N-2}\gamma_k} \cdot \mathcal{L}_{overall}^{\theta,\psi}\left(X_0,X_N\right) + \alpha \cdot \mathcal{L}_{task-related}\left(B_{N-1},Y\right).
\tag{21}
$$

# D  ALGORITHM: PSEUDO-CODE FOR TRAINING ITHP

## D.1  3-MODALITIES SCENARIO

We provide the pseudo-code for ITHP in a 3-modalities problem (please refer to Algorithm 1), which we used in our experiments including both sarcasm detection and sentiment analysis. As mentioned in C.1, we have $\left(X_0 - X_1 - X_2\right)$ as the three modal states. Here, we present the algorithm for training ITHP in a 3-modalities scenario.

## D.2  GENERAL SCENARIO FOR N MODALITIES

We additionally provide the pseudo-code and the training method for ITHP in a general form involving $N$ modalities (please refer to Algorithm 2). This broader algorithmic representation extends from the 3-modalities example delineated in D.1.

# E  MODEL ARCHITECTURE AND HYPERPARAMETERS

In this section, we provide a detailed description of our model architecture and the hyperparameter selection in different tasks for reproducibility purposes. For the ITHP, as depicted in Figure 2, to get the latent state's distribution of $B_0$, we directly use two fully connected linear layers to build the encoder, where the first layer extracts the feature vectors of input $X_0$ and the second layer outputs parameters for a latent Gaussian distribution - a mean vector and the logarithm of the variance vector. By sampling from this distribution, we obtain a latent state that represents the input state further used

---

**Algorithm 1** Pseudo-code for training ITHP in 3-modalities scenario

---

**Input:** Dataset $\mathcal{D} = \{(X_0, X_1, X_2), Y\}$; Coefficient: $\alpha, \beta, \gamma, \lambda$; Learning rate $\eta$; Mini-batch size $n_{mb}$; Assumed distribution of latent states : $\{B_0, B_1\} = \varepsilon \sim \mathcal{N}(0, 1)$;

**Output:** ITHP parameters $(\{\theta_k\}_{k=0,1}, \{\psi_k\}_{k=0,1})$; Task-related prediction $\hat{y}$;

    Initialize the parameters in the Neural Networks $(\{\theta_k\}_{k=0,1}, \{\psi_k\}_{k=0,1})$

    **repeat**

        Sample a mini-batch: $\{(X_0^n, X_1^n, X_2^n), Y^n\}_{n=1}^{n_{mb}} \sim \mathcal{D}$

        **for** $n = 1, 2, \ldots\ldots, n_{mb}$ **do**

            $\mu_0^n, \Sigma_0^n \longleftarrow f_{\theta_0}(X_0^n)$

            $z_0^n \longleftarrow \mu_0^n + \varepsilon \cdot \Sigma_0^n$

            $\hat{X}_1^n \longleftarrow f_{\psi_0}(Z_0^n)$

            $\mu_1^n, \Sigma_1^n \longleftarrow f_{\theta_1}(z_0^n)$

            $z_1^n \longleftarrow \mu_1^n + \varepsilon \cdot \Sigma_1^n$

            $\hat{X}_2^n \longleftarrow f_{\psi_1}(Z_1^n)$

        **end for**

        **if** $X_0, X_1, X_2$ are categorical data **then**

$$\mathcal{L}_{IB_0}^{\theta_0,\psi_0} = KL\left(\mathcal{N}(\mu_0^n, \Sigma_0^n) \| \mathcal{N}(0,1)\right) + \beta \cdot \frac{1}{n_{mb}} \sum_{n=1}^{n_{mb}} \left( -\sum_{C=1}^{C} X_1^n[C] \cdot \log\left(\hat{X}_1^n[C]\right) \right)$$

$$\mathcal{L}_{IB_1}^{\theta_1,\psi_1} = KL\left(\mathcal{N}(\mu_1^n, \Sigma_1^n) \| \mathcal{N}(0,1)\right) + \gamma \cdot \frac{1}{n_{mb}} \sum_{n=1}^{n_{mb}} \left( -\sum_{C=1}^{C} X_2^n[C] \cdot \log\left(\hat{X}_2^n[C]\right) \right)$$

        **else if** $X_0, X_1, X_2$ are continuous data **then**

$$\mathcal{L}_{IB_0}^{\theta_0,\psi_0} = KL\left(\mathcal{N}(\mu_0^n, \Sigma_0^n) \| \mathcal{N}(0,1)\right) + \beta \cdot \frac{1}{n_{mb}} \sum_{n=1}^{n_{mb}} \left( X_1^n - \hat{X}_1^n \right)^2$$

$$\mathcal{L}_{IB_1}^{\theta_1,\psi_1} = KL\left(\mathcal{N}(\mu_1^n, \Sigma_1^n) \| \mathcal{N}(0,1)\right) + \gamma \cdot \frac{1}{n_{mb}} \sum_{n=1}^{n_{mb}} \left( X_2^n - \hat{X}_2^n \right)^2$$

        **end if**

        $\mathcal{L}_{overall}^{\theta,\psi} = \mathcal{L}_{IB_0}^{\theta_0,\psi_0} + \lambda \cdot \mathcal{L}_{IB_1}^{\theta_1,\psi_1}$

        $\mathcal{L}_{task-related} = \frac{2}{\beta+\gamma} \cdot \mathcal{L}_{IB_{overall}}^{\theta,\psi} + \alpha \cdot \frac{1}{n_{mb}} \sum_{n=1}^{n_{mb}} (z_1^n - Y^n)^2$

        $(\theta_k, \psi_k) \longleftarrow (\theta_k, \psi_k) - \eta \cdot \nabla(\theta_k, \psi_k)\mathcal{L}_{task-related}, (k = 0, 1)$

    **until** convergence

---

to (1) feed into the multilayer perceptron (MLP) layer to predict $X_1$; (2) feed into the next hierarchical layer for another modality information. Note that this is a serial structure, and the remaining layers share the same design. The predictor is chosen specifically by the downstream tasks. For the task of sarcasm detection, we elect to use an MLP with two layers as the predictor. For the sentiment analysis task, the model ITHP-DeBERTa is trained in an end-to-end manner. The raw features of acoustic and visual modalities are pre-extracted from CMU-MOSI (Zadeh et al., 2016). The textual features are extracted by pre-trained DeBERTa (He et al., 2020; 2021). The textual features serve as input to our ITHP module while the pre-extracted acoustic and visual act as the detectors. The distilled representations that capture the information from multimodalities are generated from the ITHP module. Additionally, we incorporate residual connections for enhanced information flow. Finally, a 1-layer Transformer encoder and several fully-connected layers are designed as predictor to make sentiment predictions.

For the task of sarcasm detection, unless otherwise specified, we set the hyperparameters as follows: $\beta = 32, \gamma = 8, \lambda = 1$. We perform a 5-fold cross-validation, and for each experiment, we train the ITHP model for 200 epochs using an Adam optimizer with a learning rate of $10^{-3}$. For the task of sentiment analysis, unless otherwise specified, we set the hyperparameters as follows: $\beta = 8, \gamma = 32, \lambda = 1$. We run each experiment for 40 epochs using an Adam optimizer with a learning rate of $10^{-5}$. We repeat each experiment 5 times to calculate the mean value of the metrics. All experiments are done on an Nvidia $A$100 40GB GPUs.

## F   DATASETS DESCRIPTION AND ADDITIONAL RESULTS

By integrating DeBERTa into the highly successful fusion models of recent years, we establish new baselines for DeBERTa-based models on both the CMU-MOSI and CMU-MOSEI datasets. Our ITHP-DeBERTa model achieves state-of-the-art (SOTA) performance on both datasets. In

---

**Algorithm 2** Pseudo-code for training ITHP in general scenario

---

**Input:** Dataset $\mathcal{D} = \{(X_0, X_1, \ldots\ldots, X_N), Y\}$; Coefficient: $\alpha, \beta, \{\gamma_k\}_{k=0}^{N-2}, \{\lambda_k\}_{k=0}^{N-2}$; Learning rate $\eta$; Mini-batch size $n_{mb}$; Assumed distribution of latent states : $\{B_k\}_{k=0}^{N-1} = \varepsilon \sim \mathcal{N}(0, 1)$;

**Output:** ITHP parameters $(\{\theta_k\}_{k=0}^{N-2}, \{\psi_k\}_{k=0}^{N-2})$; Task-related prediction $\hat{y}$;

  Initialize the parameters in the Neural Networks $(\{\theta_k\}_{k=0}^{N-2}, \{\psi_k\}_{k=0}^{N-2})$

  **repeat**

    Sample a mini-batch: $\{(X_0^n, X_1^n, \ldots\ldots, X_N^n), Y^n\}_{n=1}^{n_{mb}} \sim \mathcal{D}$

    **for** $n = 1, 2, \ldots\ldots, n_{mb}$ **do**

      $\mu_0^n, \Sigma_0^n \longleftarrow f_{\theta_0}(X_0^n)$

      $z_0^n \longleftarrow \mu_0^n + \varepsilon \cdot \Sigma_0^n$

      $\hat{X}_1^n \longleftarrow f_{\psi_0}(Z_0^n)$

      **for** $k = 0, 1, 2, \ldots\ldots, N-2$ **do**

        $\mu_{k+1}^n, \Sigma_{k+1}^n \longleftarrow f_{\theta_{k+1}}(z_k^n)$

        $z_{k+1}^n \longleftarrow \mu_{k+1}^n + \varepsilon \cdot \Sigma_{k+1}^n$

        $\hat{X}_{k+2}^n \longleftarrow f_{\psi_{k+1}}(Z_{k+1}^n)$

      **end for**

    **end for**

    **if** $X_0, X_1, \ldots\ldots, X_N$ are categorical data **then**

      $\mathcal{L}_{IB_0}^{\theta_0, \psi_0} = KL\left(\mathcal{N}(\mu_0^n, \Sigma_0^n) \| \mathcal{N}(0,1)\right) + \beta \cdot \frac{1}{n_{mb}} \sum_{n=1}^{n_{mb}} \left(-\sum_{C=1}^{C} X_1^n[C] \cdot \log\left(\hat{X}_1^n[C]\right)\right)$

      **for** $k = 0, 1, 2, \ldots\ldots, N-2$ **do**

        $\mathcal{L}_{IB_{k+1}}^{\theta_{k+1}, \psi_{k+1}} = KL\left(\mathcal{N}(\mu_{k+1}^n, \Sigma_{k+1}^n) \| \mathcal{N}(0,1)\right) +$

        $\gamma_k \cdot \frac{1}{n_{mb}} \sum_{n=1}^{n_{mb}} \left(-\sum_{C=1}^{C} X_{k+2}^n[C] \cdot \log\left(\hat{X}_{k+2}^n[C]\right)\right)$

      **end for**

    **else if** $X_0, X_1, \ldots\ldots, X_N$ are continuous data **then**

      $\mathcal{L}_{IB_0}^{\theta_0, \psi_0} = KL\left(\mathcal{N}(\mu_0^n, \Sigma_0^n) \| \mathcal{N}(0,1)\right) + \beta \cdot \frac{1}{n_{mb}} \sum_{n=1}^{n_{mb}} \left(X_1^n - \hat{X}_1^n\right)^2$

      **for** $k = 0, 1, 2, \ldots\ldots, N-2$ **do**

        $\mathcal{L}_{IB_{k+1}}^{\theta_{k+1}, \psi_{k+1}} = KL\left(\mathcal{N}(\mu_{k+1}^n, \Sigma_{k+1}^n) \| \mathcal{N}(0,1)\right) + \gamma_k \cdot \frac{1}{n_{mb}} \sum_{n=1}^{n_{mb}} \left(X_{k+2}^n - \hat{X}_{k+2}^n\right)^2$

      **end for**

    **end if**

    $\mathcal{L}_{overall}^{\theta, \psi} = \mathcal{L}_{IB_0}^{\theta_0, \psi_0} + \sum_{k=0}^{N-2} \lambda_k \cdot \mathcal{L}_{IB_{k+1}}^{\theta_{k+1}, \psi_{k+1}}$

    $\mathcal{L}_{task-related} = \frac{N-1}{\beta + \sum_{k=0}^{N-2} \gamma_k} \cdot \mathcal{L}_{IB_{overall}}^{\theta, \psi} + \alpha \cdot \frac{1}{n_{mb}} \sum_{n=1}^{n_{mb}} \left(z_{k+1}^n - Y^n\right)^2$

    $(\theta_k, \psi_k) \longleftarrow (\theta_k, \psi_k) - \eta \cdot \bigtriangledown(\theta_k, \psi_k) \mathcal{L}_{task-related}, (k = 0, 1, 2, \ldots\ldots, N-2)$

  **until** convergence

---

the case of Self-MM, we utilize the raw data and codebase provided in the self-mm repository [https://github.com/thuiar/Self-MM, MIT License] and modify the feature extraction method to employ DeBERTa, replacing the original requirement for BERT. However, we note that Self-MM itself heavily relies on the feature extraction process performed by BERT, resulting in a significant degradation of DeBERTa-based Self-MM, which restricts the applicability of Self-MM in certain scenarios.

## F.1 CMU-MOSI

The MOSI dataset comprises 2,199 video clips from YouTube movie reviews, annotated with sentiment intensity on a -3 to +3 scale. This multimodal dataset integrates transcriptions, visual gestures, and audio-visual features, facilitating comprehensive sentiment analysis research. Its unique structuring allows nuanced sentiment interpretation at opinion level Gandhi et al. (2022).

| Task Metric | BA↑ | F1↑ | MAE↓ | Corr↑ |
|---|---|---|---|---|
| **BERT** | | | | |
| TFN (Zadeh et al., 2017) | 74.8 | 74.1 | 0.955 | 0.649 |
| MARN (Zadeh et al., 2018c) | 77.7 | 77.9 | 0.938 | 0.691 |
| MFN (Zadeh et al., 2018a) | 78.2 | 78.1 | 0.911 | 0.699 |
| RMFN (Liang et al., 2018) | 79.6 | 78.9 | 0.878 | 0.712 |
| LMF (Liu et al., 2018) | 79.1 | 77.3 | 0.899 | 0.701 |
| MulT (Tsai et al., 2019) | 81.5 | 80.6 | 0.861 | 0.711 |
| Self-MM$_b$ (Yu et al., 2021) | 84.0 | 84.4 | 0.713 | 0.798 |
| MMIM$_b$ (Han et al., 2021b) | 84.1 | 84.0 | 0.700 | 0.800 |
| MAG$_b$ (Rahman et al., 2020) | 84.2 | 84.1 | 0.712 | 0.796 |
| **DeBERTa** | | | | |
| MAG$_d$ | 86.1 | 86.0 | 0.690 | 0.831 |
| **ITHP** | **88.7** | **88.6** | **0.643** | **0.852** |
| Human | 85.7 | 87.5 | 0.710 | 0.820 |

Table 4: **Model performance on CMU-MOSI.** The table presents the performance of different BERT-based pre-trained models. Models developed by us are highlighted in bold, and optimal results are underlined. ITHP$_d$ consistently outperforms other models. The metrics used include Binary Accuracy (BA), F1 Score (F1), Mean Absolute Error (MAE), and Pearson Correlation (Corr).

| Task Metric | BA↑ | F1↑ | MAE↓ | Corr↑ |
|---|---|---|---|---|
| **BERT** | | | | |
| Self-MM$_b$ (Han et al., 2021b) | 85.0 | 85.0 | 0.529 | 0.767 |
| MMIM$_b$ (Han et al., 2021b) | 86.0 | 86.0 | 0.526 | 0.772 |
| MAG$_b$ (Han et al., 2021b) | 84.8 | 84.7 | 0.543 | 0.755 |
| **DeBERTa** | | | | |
| Self-MM$_d$ | 65.3 | 65.4 | 0.813 | 0.208 |
| MMIM$_d$ | 85.2 | 85.4 | 0.568 | 0.799 |
| MAG$_d$ | 85.8 | 85.9 | 0.636 | 0.800 |
| **ITHP** | **87.3** | **87.4** | **0.564** | **0.813** |

Table 5: **Model performance on CMU-MOSEI dataset.** Models developed by us are highlighted in bold, and optimal results are underlined. The metrics used include Binary Accuracy (BA), F1 Score (F1), Mean Absolute Error (MAE), and Pearson Correlation (Corr).

## F.2 CMU-MOSEI

The CMU Multimodal Opinion Sentiment and Emotion Intensity (CMU-MOSEI) dataset stands as the most extensive dataset available for multimodal sentiment analysis and emotion recognition. It encompasses over 23,500 sentence utterance videos derived from more than 1000 YouTube speakers, ensuring a diverse and gender-balanced collection. The sentence utterances, sourced from a variety of topics and monologue videos, are randomly selected, transcribed, and appropriately punctuated. This dataset represents an expanded version of MOSI, which contained 3,228 videos and 23,453 utterances, offering a richer resource for research and development in sentiment analysis and emotion recognition.

## F.3 MUStARD

MUStARD is a dataset constructed to support the exploration of automated sarcasm detection. Comprised of 690 annotated video utterances from various TV shows like Friends and The Big Bang Theory, it offers a diverse range of sarcastic and non-sarcastic instances. The utterances, spanning video, audio, and text modalities, are each associated with speaker identifiers and contextual information.

We benchmarked against another state-of-the-art (SOTA) Hierarchical Fusion Model (HFM) (Cai et al., 2019b), which employs a task-oriented hierarchical fusion approach. This approach integrates early fusion, representation fusion, and modality fusion techniques to augment the understanding and representation of individual modalities. To maintain consistency in our comparative framework, we integrated HFM's modality fusion module into our experimental setup. The subsequent tests on our dataset yielded results that facilitate a direct performance comparison between HFM and our proposed model.

| Model | Precision | Recall | F-Score |
|-------|-----------|--------|---------|
| MSDM  | 71.9      | 71.4   | 71.5    |
| HFM   | 69.4      | 68.8   | 68.9    |
| ITHP  | **75.3**  | **75.2** | **75.2** |

Table 6: **Results of sarcasm detection on the MUStARD dataset.** This table displays the weighted Precision, Recall, and F-Score for the V-T-A (video, text, audio) modality, averaged across five folds. Bold figures represent the highest scores achieved.

## F.4 VARYING HYPERPARAMETERS

The sizes of the latent states ($B_0$ and $B_1$) have an impact on the performance of the model. The small size may not capture sufficient information, whereas a large size may introduce redundancy and hinder the goal of compactness. To investigate the impact of different sizes, we vary the dimensions of the latent states and evaluate the model's performance. The results are depicted in Fig. 6.

In Table 6, we present the comparative results of ITHP, MSDM, and HFM for sarcasm detection on the MUStARD dataset. For the HFM model, we adopted the Modality Fusion structure mentioned in its original publication to facilitate multimodal integration. As stated in the HFM paper, the modalities they employed were image, attribute, and text modalities. In our implementation, these were substituted with video, audio, and text modalities, corresponding to the MUStARD dataset's specifications. Regarding parameter settings, we retained all original configurations of the HFM model, making adjustments only to the input channel numbers of certain layers to align with the input data requirements of the MUStARD dataset.

The results indicate that when the sizes of $B_0$ and $B_1$ are both set to 8, the model exhibits the poorest performance, suggesting that this size may be insufficient for effectively capturing and conveying the information. On the other hand, when $B_0$ size is set to 128 and $B_1$ size is set to 64, we observe optimal performance with a weighted precision of 0.754. This finding suggests that a latent space dimension of 64 is adequate for representing the multimodal information, while the dimensions of the embedding features of the modalities are $d_v = 2048$, $d_t = 768$, and $d_a = 283$.

The specific values of the weighted precision and recall shown in Fig. 4 are provided in Table 7 and Table 8, respectively. Similarly, the specific values of the weighted precision and recall shown in Fig. 6 are presented in Table 9 and Table 10, respectively.

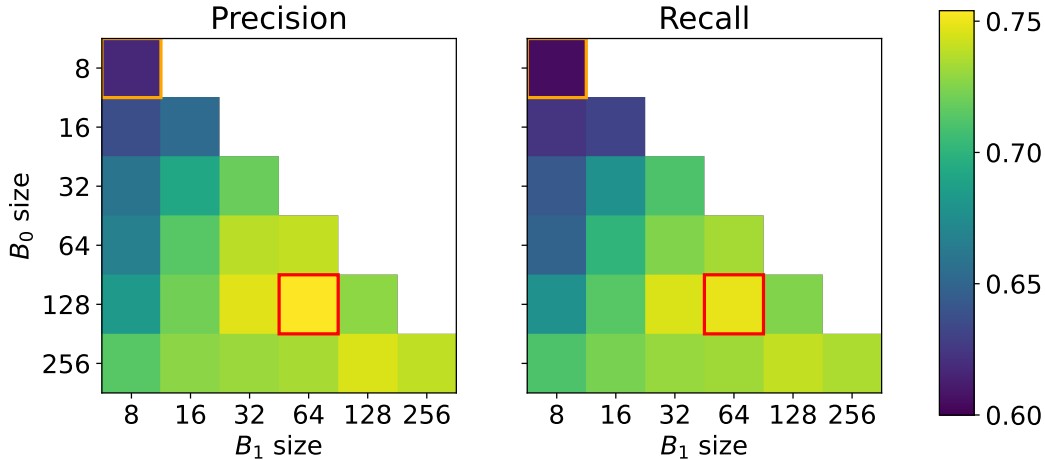

Figure 6: **Weighted precision and recall for the binary classification task under varying sizes of latent states.** The graph shows the impact of varying the sizes of latent states ($B_0$ and $B_1$). For each figure, the Red (Orange) color denotes the highest (lowest) value, respectively.

**Precision**

| $\gamma$ \ $\beta$ | 2 | 4 | 8 | 16 | 32 | 64 |
|---|---|---|---|---|---|---|
| 2 | 0.619 | 0.632 | 0.702 | 0.735 | 0.742 | 0.739 |
| 4 | 0.621 | 0.655 | 0.715 | 0.742 | 0.749 | 0.744 |
| 8 | 0.637 | 0.658 | 0.734 | 0.745 | 0.753 | 0.746 |
| 16 | 0.693 | 0.686 | 0.736 | 0.746 | 0.749 | 0.740 |
| 32 | 0.705 | 0.714 | 0.734 | 0.747 | 0.746 | 0.748 |
| 64 | 0.710 | 0.728 | 0.729 | 0.743 | 0.747 | 0.741 |

Table 7: **Weighted precision for the binary classification task under varying Lagrange multipliers.** The table shows the corresponding values of weighted precision of Lagrange multipliers ($\beta$ and $\gamma$). The optimal result is underlined.

**Recall**

| $\gamma$ \ $\beta$ | 2 | 4 | 8 | 16 | 32 | 64 |
|---|---|---|---|---|---|---|
| 2 | 0.607 | 0.627 | 0.696 | 0.737 | 0.742 | 0.736 |
| 4 | 0.619 | 0.650 | 0.706 | 0.740 | 0.747 | 0.742 |
| 8 | 0.622 | 0.649 | 0.736 | 0.742 | 0.752 | 0.742 |
| 16 | 0.681 | 0.672 | 0.726 | 0.745 | 0.747 | 0.735 |
| 32 | 0.692 | 0.698 | 0.722 | 0.745 | 0.741 | 0.748 |
| 64 | 0.703 | 0.721 | 0.723 | 0.740 | 0.745 | 0.738 |

Table 8: **Weighted recall for the binary classification task under varying Lagrange multipliers.** The table shows the corresponding values of weighted recall for different combinations of Lagrange multipliers ($\beta$ and $\gamma$). The optimal result is underlined.

| | **Precision** | | | | | |
|---|---|---|---|---|---|---|
| $B_1$ size / $B_0$ size | **8** | **16** | **32** | **64** | **128** | **256** |
| **8** | 0.617 | - | - | - | - | - |
| **16** | 0.637 | 0.654 | - | - | - | - |
| **32** | 0.659 | 0.692 | 0.719 | - | - | - |
| **64** | 0.667 | 0.714 | 0.738 | 0.740 | - | - |
| **128** | 0.683 | 0.721 | 0.747 | 0.754 | 0.728 | - |
| **236** | 0.714 | 0.727 | 0.731 | 0.734 | 0.745 | 0.739 |

Table 9: **Weighted precision for the binary classification task under varying sizes of latent states $B_0$ and $B_1$.** The table shows the corresponding values of weighted precision for different combinations of $B_0$ and $B_1$. The optimal result is underlined.

| | **Recall** | | | | | |
|---|---|---|---|---|---|---|
| $B_1$ size / $B_0$ size | **8** | **16** | **32** | **64** | **128** | **256** |
| **8** | 0.605 | - | - | - | - | - |
| **16** | 0.624 | 0.631 | - | - | - | - |
| **32** | 0.643 | 0.678 | 0.711 | - | - | - |
| **64** | 0.649 | 0.701 | 0.725 | 0.733 | - | - |
| **128** | 0.678 | 0.714 | 0.745 | 0.749 | 0.725 | - |
| **236** | 0.711 | 0.723 | 0.730 | 0.732 | 0.740 | 0.735 |

Table 10: **Weighted recall for the binary classification task under varying sizes of latent states $B_0$ and $B_1$.** The table shows the corresponding values of weighted recall for different combinations of $B_0$ and $B_1$. The optimal result is underlined.

## F.5 VARYING ORDERS OF MODALITIES

Our research includes varying the order of modalities to assess the impact on our model's performance. The detailed results are catalogued in Table 11, which shows the effects of different modality sequences on weighted precision, recall, and F-score.

The results reveals that the modality sequence significantly affects performance metrics. Specifically, positioning video as the primary modality (V→T→A) yielded the highest scores, with all metrics around 75%. In contrast, sequences starting with audio (A→T→V) registered the lowest, with scores near 71%. This evidence underscores the importance of strategic modality ordering, which can be guided by domain expertise to enhance model efficiency for specific tasks.

| Metrics | **Order of Modalities** | | | | | |
|---|---|---|---|---|---|---|
| | **V→T→A** | **V→A→T** | **T→V→A** | **T→A→V** | **A→V→T** | **A→T→V** |
| Precision | 75.3 | 73.7 | 73.0 | 72.0 | 71.4 | 70.8 |
| Recall | 75.2 | 73.5 | 72.8 | 71.8 | 71.2 | 70.6 |
| F-Score | 75.2 | 73.4 | 72.9 | 71.7 | 71.5 | 70.6 |

Table 11: **Varying Orders of Modalities.** The table showcases the impact of changing the order of modalities on the weighted Precision, Recall, and F-Score across both sarcastic and non-sarcastic classes, averaged across five folds. Underlined values highlight the best results for each metric. The modalities are denoted as: $T$ for text, $A$ for audio, and $V$ for video.

## G  ASSESSMENT OF INFERENCE LATENCY

Our study included a comparative analysis of inference latency, utilizing a simplified multi-layer perceptron (MLP) as a benchmark. The MLP head, consisting of 64 neurons, served as the predictor within our ITHP model framework. In this setup, we concatenated modalities M0, M1, and M2 and subjected them to the MLP head. The resulting inference latency was approximately 0.00221 milliseconds (ms) per sample. By comparison, the complete ITHP model, which features a more complex hierarchical structure, exhibited an inference latency of 0.0120 ms per sample.

It is important to note that the detectors within our ITHP model remain inactive during the inference phase, which contributes to the model's efficiency. Despite the inherent latency increment attributable to the hierarchical design, the ITHP model demonstrates commendable performance efficiency. The observed slight increase in latency is a testament to the model's capability to balance the demands of complex multimodal learning tasks with the necessity for swift processing times.

## H  LEARNING COMPLEMENTARY INFORMATION FROM SECONDARY MODALITIES

Here, we provide additional elaboration on the potential limitations that may arise when the primary modality lacks sufficient information. It's essential to highlight that our selection criteria for determining the primary modality have been designed to encapsulate the most pertinent information. However, it's equally vital to acknowledge that depending solely on the primary modality might not always suffice, necessitating the inclusion of information from secondary modalities. By harnessing the capabilities of our deep learning framework, our model has a certain ability to learn the complementary information from auxiliary modalities. To achieve a harmonious representation, we've incorporated specific terms in the loss function. Specifically, taking the 3-modalities learning problem in Eqn. (3) as an example, term $L_1 = -\beta I(B_0; X_1)$ is tasked with supervising the knowledge transfer from $X_1$ to $B_0$, while term $L_2 = -\gamma I(B_1; X_2)$ is responsible for supervising the knowledge dissemination from $X_2$ to $B_1$. Through this framework, the ITHP model is primed to discern and incorporate complementary information from non-input modalities. Our model, therefore, endeavors to strike a balance between mutual information, irrelevant data, and task-essential information, ensuring an optimized flow of information.

## I  METRICS FOR EVALUATION

We employ a variety of metrics for evaluating the performance of models in binary classification: weighted precision for different tasks. These metrics include weighted precision, weighted recall, weighted F score, Binary Accuracy (BA), F1-Score (F1), Mean Absolute Error (MAE), and Pearson Correlation (Corr). We consider the following notations: $TP$ (True Positive); $TN$ (True Negative); $FP$ (False Positive); and $FN$ (False Negative).

### I.1  SARCASM DETECTION

To ensure a fair comparison with the model performance outlined in (Castro et al., 2019), we have utilized the same evaluation metrics: weighted precision, weighted recall, and weighted F-score, following the guidelines provided by (Castro et al., 2019). The weighted average precision, recall, and F-score are determined by incorporating the support (number of true instances for each label) as weights. The support for class $i$ is denoted as $S_i$. Here are the calculations for these metrics:

**Weighted Precision** $P_w$**:** It measures the proportion of correct positive predictions, adjusting for class prevalence. The formula is given as:

$$P_w = \frac{\sum_{i=1}^{n}(S_i * precision_i)}{\sum_{i=1}^{n} S_i}, \tag{22}$$

where $precision_i = \frac{TP_i}{(TP_i + FP_i)}$.

**Weighted Recall** $R_w$**:** This metric estimates the model's ability to identify all positive instances, accounting for the frequency of each class. It is computed as:

$$R_w = \frac{\sum_{i=1}^{n}(S_i * recall_i)}{\sum_{i=1}^{n} S_i}, \tag{23}$$

where $recall_i = \frac{TP_i}{(TP_i + FN_i)}$.

**Weighted F-score** $F_w$**:** This balances precision and recall, weighting based on the true instances for each label. The formula for this metric is:

$$F_w = \frac{\sum_{i=1}^{n}(S_i * F_i)}{\sum_{i=1}^{n} S_i}, \tag{24}$$

where $F_i = \frac{2*(precision_i * recall_i)}{(precision_i + recall_i)}$.

In these formulas, $TP_i$ represents the true positives for class $i$, $FP_i$ represents the false positives for class $i$, $FN_i$ represents the false negatives for class $i$, and $S_i$ represents the number of true instances for each label (class $i$).

## I.2 SENTIMENT ANALYSIS

We adopt the well-acknowledged metrics in sentiment analysis, specifically, Binary Accuracy (BA), F1-Score (F1), Mean Absolute Error (MAE), and Pearson Correlation (Corr), to evaluate our model's performance and contrast it with the benchmark models on CMU-MOSI and CMU-MOSEI datasets. The formulas for these metrics are as follows:

**Binary Accuracy (BA):** It's a straightforward measure calculated as the number of correct predictions divided by the total number of predictions. Here's the formula:

$$BA = \frac{TP + TN}{TP + TN + FP + FN}. \tag{25}$$

**F1-Score (F1):** The F1 Score is a metric that seeks to strike a balance between Precision and Recall. Here's the formula:

$$F1 = 2 * \frac{(Precision * Recall)}{(Precision + Recall)}, \tag{26}$$

where $Precision = \frac{TP}{TP+FP}$ and $Recall = \frac{TP}{TP+FN}$.

**Mean Absolute Error (MAE):** The MAE quantifies the average magnitude of errors in a set of predictions, disregarding their direction. Here's the formula:

$$\text{MAE}(\hat{y}, y) = \frac{\sum_{i=1}^{n} |\hat{y}_i - y_i|}{n}, \tag{27}$$

where $y_i$ is the true label, $\hat{y}_i$ is the predictive value and $n$ is the total number of predictions.

**Pearson Correlation (Corr):** Pearson Correlation (Corr) assesses the linear association between predicted probabilities and true labels. Here's the formula:

$$\text{Corr}(x, y) = \frac{\sum_{i=1}^{n} (x_i - \bar{x})(y_i - \bar{y})}{\sqrt{\sum_{i=1}^{n} (x_i - \bar{x})^2 \sum_{i=1}^{n} (y_i - \bar{y})^2}}, \tag{28}$$

where $x_i$ refers to predicted probabilities for the positive class, and $y_i$ is the true label. The $\bar{x}$ and $\bar{y}$ are the means of the predicted probabilities and actual labels, respectively.

## J ALGORITHMS FOR RANKING MODALITIES

### J.1 ADAPTING SAMPLE ENTROPY (SAMPEN) FOR DATASETS

Sample Entropy (SampEn) is a method traditionally used for time-series data to quantify their complexity by evaluating the predictability of patterns within the data. When adapting SampEn for

feature datasets, especially those resembling time-series data in structure, the same principles are used to analyze the distribution and regularity of features within individual samples in a multidimensional feature space.

For a dataset represented as a set of feature vectors, where each vector can be considered as a sequence, the information richness can be quantified by measuring the regularity and unpredictability of these sequences. SampEn in this context measures the likelihood that subsequences of features within a given vector that are close (within a certain tolerance) remain close when one additional dimension is considered. This approach evaluates the diversity and distribution of the patterns within each individual sample, reflecting the intrinsic information richness of the feature data.

The SampEn of a dataset is calculated as follows:

1. Choose an embedding dimension $m = 2$ and a tolerance $r = 0.2 \times$ standard deviation of the data.

2. For each sample in the dataset, form vectors $U_i^m$ for $i = 1, \ldots, D - m$ where $D$ is the number of features.

3. Define $B^m(r)$ as the count of times when two vectors $U_i^m$ and $U_{i+1}^m$ within the same sample are within $r$ of each other.

4. Define $A^m(r)$ similarly for $m + 1$.

5. SampEn is given by:
$$\text{SampEn}(m, r, N) = -\ln\left(\frac{A^m(r)}{B^m(r)}\right)$$

---

**Algorithm 3** Ranking Modalities Based on SampEn

---

**function** SAMPLEENTROPY($data, m, r$)
    $N \leftarrow$ number of samples in $data$
    $B^m \leftarrow 0$
    $A^m \leftarrow 0$
    **for** each sample in data **do**
        **for** $i \leftarrow 1$ to number of features $- m$ **do**
            **if** norm(sample$[i : i + m]$ − sample$[i + 1 : i + m + 1]$) $< r$ **then**
                $B^m \leftarrow B^m + 1$
                **if** $i <$ number of features $- m - 1$ and norm(sample$[i : i + m + 1]$ − sample$[i + 1 : i + m + 2]$) $< r$ **then**
                    $A^m \leftarrow A^m + 1$
                **end if**
            **end if**
        **end for**
    **end for**
    **return** $- \log(A^m / B^m)$         ▷ SampEn value
**end function**

**function** RANKMODALITIES($modalities, m, r$)
    $ModalityList \leftarrow [\ ]$
    **for** each $modality$ in $modalities$ **do**
        $entropy \leftarrow$ SAMPLEENTROPY($modality, m, r$)
        Append $(modality, entropy)$ to $ModalityList$
    **end for**
    Sort $ModalityList$ based on entropy in descending order
    **return** $ModalityList$         ▷ List of modalities ranked by entropy
**end function**

---

## J.2 RANKING MODALITIES USING SUBMODULAR FUNCTIONS

Given a set of $N$ different modalities $M = \{m_1, m_2, \ldots, m_N\}$, the goal is to find a subset of modalities $S \subseteq M$ that maximizes the predictive performance in a given downstream task.

**Submodular Function Definition:** Let $f : 2^M \to \mathbb{R}$ be a submodular function that maps a set of modalities to a real number indicating their performance in a predictive task. For any $A \subseteq B \subseteq M$ and $x \notin B$, the function $f$ satisfies:

$$f(A \cup \{x\}) - f(A) \geq f(B \cup \{x\}) - f(B)$$

**Greedy Algorithm:** The greedy algorithm starts with an empty set and iteratively adds a modality that maximizes the function $f(S)$:

$$S_{i+1} = S_i \cup \{\arg\max_{x \notin S_i} f(S_i \cup \{x\})\}$$

**Performance Guarantee:** The greedy algorithm provides an approximation guarantee of $(1 - \frac{1}{e})$ for maximizing a submodular function, where $e$ is the base of the natural logarithm.

Assuming a function '$evaluatePerformance$' that assesses the performance of a given set of modalities.

---

**Algorithm 4** Greedy Algorithm for Ranking Modalities

---

**function** GREEDYSUBMODULARSORT($modalities, evaluatePerformance$)
    $S \leftarrow$ empty set
    $ModalityList \leftarrow$ empty list
    **while** $|S| < |modalities|$ **do**
        $bestImprovement \leftarrow -\infty$
        $bestModality \leftarrow$ null
        **for each** $modality$ in modalities $- S$ **do**
            $newPerformance \leftarrow evaluatePerformance(S \cup \{modality\})$
            $improvement \leftarrow newPerformance - evaluatePerformance(S)$
            **if** $improvement > bestImprovement$ **then**
                $bestImprovement \leftarrow improvement$
                $bestModality \leftarrow modality$
            **end if**
        **end for**
        **if** $bestModality$ is null **then**
            **break**
        **end if**
        $S \leftarrow S \cup \{bestModality\}$
        $ModalityList.\text{append}(bestModality)$
    **end while**
    **return** $ModalityList$
**end function**

---

## K  DEFAULT NOTATION

| | |
|---|---|
| $X_0$ | The input modal state with the highest information richness |
| $\{X_k\}_{k=1}^N$ | The other modal states for information fusion |
| $\{B_k\}_{k=0}^{N-1}$ | Latent states constructed for ITHP |
| $Y$ | Task-related labels or values |
| $d_{X_k}/d_{B_k}$ | The dimensions for modal states/ latent states |
| $p(\cdot)$ | Deterministic distribution of random variables |
| $q(\cdot)$ | Random probability distribution fitted by the neural network |
| $N$ | Total number of modal states |
| $\beta$ | Lagrange multiplier for the first level IB |
| $\gamma_k$ | Lagrange multipliers for the subsequent level IBs |
| $\lambda_k$ | Coefficients for integrating the first and sebsequent level IBs |
| $\alpha$ | Coefficient for balancing the ITHP loss and the task-related loss |
| $\theta_k$ | Parameters for the encoding-level Neural Networks |
| $\psi_k$ | Parameters for the predicting-level Neural Networks |
| $\mu_k$ | Mean value learned by the encoding neural networks |
| $\Sigma_k$ | Covariance value learned by the encoding neural networks |
| $\varepsilon$ | Assumed distribution of latent states |
| $n_{mb}$ | Mini-batch size for training the ITHP |
| $\eta$ | Learning rate for training the ITHP |

