# OpenReview forum: "Neuro-Inspired Information-Theoretic Hierarchical Perception for Multimodal Learning"
_ICLR.cc/2024/Conference — ICLR 2024 poster_

### Official Review · Reviewer_JgMF · 2023-10-29

**Soundness:** 3 good
**Presentation:** 3 good
**Contribution:** 3 good
**Rating:** 6
**Confidence:** 3

**Summary:**

This paper proposes an Information-Theoretic Hierarchical Perception (ITHP) model, based on the concept of information bottleneck, for effective multimodal fusion.

**Strengths:**

+ The proposed Information-Theoretic Hierarchical Perception model looks novel to me; This stands in contrast to mainstream multimodal fusion where different modalities are treated equally, and the final prediction is usually acquired by concatenating the features from each modality. The concept of designating a prime modality and using the other modalities to guide the flow of information looks interesting, and makes sense to me --- In many multimodal tasks, modalities do not contribute equally to the final prediction.

+ The experimental results are strong. The analysis about varying Lagrange multiplier in Sec. 3.1 provides good insight.

+ Many design choices (e.g. latent state size, order of modalities) are thoroughly evaluated in the appendix.

**Weaknesses:**

+ As the authors already point out in Section 5, the proposed approach requires a predefined order of modalities, which could be a problem when there are > 3 modalities and we do not have prior information about the multimodal task.

+ Would the proposed hierarchical architecture introduce some inference latency compared with the standard non-hierarchical approaches (Take Figure 3 as an example, say compared to concatenation of $X_0$, $X_1$ and $X_2$ and then passed to an MLP head)?

+ It seems in the current framework, feature extraction from raw video, audio & text & the proposed ITHP are separate (first feature extraction then use the encoded features in the hierarchical information flow). The temporal nature of these data is not utilized. I wonder if the authors could briefly comment on the potential of extending the ITHP model to handle sequence data (that naturally comes with a temporal order).

**Questions:**

See weaknesses.

---

> ### Author Response · Authors · 2023-11-21
> **Response to Reviewer JgMF**
>
> Thank you for acknowledging the novelty and sensibility of our Information-Theoretic Hierarchical Perception (ITHP) model, as well as for recognizing the strength of our experimental results. We also greatly value your insightful comments, which have guided us in further enhancing our paper. Here  we provide the point-wise response to your concerns:
>
> **Algorithms for defining the order of modalities:** Thank you for pointing out your concern on defining the order of the modalities when there are > 3 modalities and we do not have prior information about the multimodal task. The sequence of modalities indeed influences the performance of our model. When the number of modalities increases and there is a lack of relevant prior knowledge, determining the richness of information in each modality and the sequence among them becomes crucial. To address this problem, we have developed practical algorithms to define the order of modalities: 1. We employ Sample Entropy (SampEn) to assess the information richness within different modalities, which serves as the basis for ranking these modalities. 2. For specific downstream tasks, we implement a greedy algorithm in conjunction with a submodular function to more comprehensively determine the order of the modalities. Detailed explanations and algorithms for these methods are provided in the appendix J (page 26-27), as also mentioned in the 'Response to all reviewers' comments above.
>
> **Inference latency:** Thank you for your valuable suggestion to test the inference latency, which has enabled a more thorough assessment of our model's performance. In response to your concern about the inference latency of our proposed hierarchical architecture as compared to standard non-hierarchical approaches, we conducted a comparative analysis using a simple MLP head. This involved concatenating modalities M0, M1, and M2 and processing them through an MLP head, representative of the predictor component in our ITHP model. The inference latency for this simple configuration was approximately 0.00221 ms per sample. In contrast, our full ITHP model, incorporating a hierarchical structure, recorded an inference latency of 0.0120 ms per sample using the same experiment setting. It's noteworthy that during inference, the detectors in our model are not active, contributing to its overall efficiency. Despite the anticipated increase in latency compared to an MLP head due to its hierarchical nature, our model remains competitive in terms of efficiency. We appreciate your suggestion and have included the above-mentioned discussion in Appendix G (page 24).

---

> > ### Author Response · Authors · 2023-11-21
> > **Continue 1**
> >
> > **The potential of extending the ITHP model to handle sequence data:**
> >
> > Thank you for your insightful comment and question on the potential extension of the ITHP model to handle sequence data and the non-stationary properties of temporal data.
> >
> > Currently, our ITHP framework separates feature extraction from the hierarchical information flow. As per your suggestion, we see the value in integrating sequence handling directly into the ITHP model. Because the ITHP design is compatible with existing neural network architectures, it could be merged with models such as LSTM [1], Transformers [2], etc which excel in capturing temporal dynamics crucial in multimodal learning scenarios. In this setup, we could leverage such a neural network to process the sequence inputs in the prime modality while the ITHP model integrates this temporal information with other modalities hierarchically. This approach would allow us to effectively utilize the temporal nature of the data.
> >
> > Simultaneously, we acknowledge the non-stationary nature [3] of temporal data where the statistical properties of the data can change over time. In response to this challenge, we could consider each distinct statistical state in the non-stationary temporal data as a separate modality. By treating these states as separate modalities, we could apply the ITHP model to discover the shared underlying causal relationships between these states, allowing us to effectively uncover dynamic interactions and dependencies between different temporal states.
> >
> > This extension of the ITHP model would involve identifying changes in the statistical properties of the temporal data, segmenting the data based on these changes, and then applying the ITHP model to this segmented data. In doing so, we could potentially capture the non-stationary characteristics of the temporal data and improve the model's performance in tasks that involve such data.
> >
> > Your comment has opened an exciting avenue for future work. We appreciate your thoughtful feedback and will consider it as we continue to refine and extend the ITHP model. We are eager to explore how the ITHP model can be adapted to handle the unique challenges and opportunities presented by non-stationary temporal data.
> >
> > [1] Cheng, Fei, and Yusuke Miyao. "Classifying temporal relations by bidirectional LSTM over dependency paths." Proceedings of the 55th Annual Meeting of the Association for Computational Linguistics (Volume 2: Short Papers). 2017.
> >
> > [2] Zhou, Haoyi, et al. "Informer: Beyond efficient transformer for long sequence time-series forecasting." Proceedings of the AAAI conference on artificial intelligence. Vol. 35. No. 12. 2021.
> >
> > [3] Huang, Biwei, Kun Zhang, Jiji Zhang, Joseph Ramsey, Ruben Sanchez-Romero, Clark Glymour, and Bernhard Schölkopf. "Causal discovery from heterogeneous/nonstationary data." The Journal of Machine Learning Research 21, no. 1 (2020): 3482-3534.

---

### Official Review · Reviewer_4bfy · 2023-10-30

**Soundness:** 2 fair
**Presentation:** 3 good
**Contribution:** 2 fair
**Rating:** 6
**Confidence:** 3

**Summary:**

The paper proposes a neuro-inspired sequential and hierarchical processing model for multi-modality data. The key is to balance the minimization of mutual information between the latent state and the input modal state, and the maximization of mutual information between the latent states and the remaining modal states. The paper presents the theory formulation of the processing steps and evaluate on two multimodality tasks, sarcasm detection and sentiment analysis.

**Strengths:**

1.	The paper provided detailed formulation of the perception process and deduction of the loss function.
2.	The paper conducted detailed experiments on the ablation of hyperparameters.

**Weaknesses:**

1.	The inspiration from neuroscience research seems simple, and unnecessary to the main processing pipeline of the main work, making the main work distracting.
2.	The order of the modality in processing the data seems quite important, which highly impacts the performance. For such an importance factor, it is better to provide further theoretical analysis or practical guidance on it.
3.	The paper should include some recent SOTA works, at lease works in 2022, on comparing the sentiment analysis.

**Questions:**

1.	Why sequential processing is good? Except for inspiration from neural inspirations. This operation introduces the extra problem of the order of modality.
2.	Does the best beta and gamma share across different tasks? Or it is needed to balance for different tasks, making this method quite cumbersome.

---

> ### Author Response · Authors · 2023-11-21
> **Response to Reviewer 4bfy**
>
> Thank you for your thorough review of our paper, we greatly value your insightful comments. Your suggestions have helped us enhance the content of our paper. Here we provide our responses to the concerns you raised.
>
> **Regarding to the inspiration from neuroscience research:** Our model is inspired by neuroscience research, highlighting how the brain hierarchically processes multimodal information. We have integrated an information bottleneck framework into our design to emulate the hierarchical process, which stands as an innovation in our work and sets our model apart from traditional approaches. Rather than initiating with a fusion structure as is common in conventional models, our model adopts a comprehensive, information theoretic approach.
>
> **Sequential processing and order of modalities:** We are grateful for your recommendations regarding offering theoretical analysis or practical guidance for determining the order of modalities. Based on your advice and comments from other reviewers, we believe it is necessary to provide practical guidance for defining the order of modalities. We have developed practical algorithms to define the primary modality and the order of modalities: (i) We employ Sample Entropy (SampEn) to assess the information richness within different modalities, which serves as the basis for ranking these modalities. (ii) For specific downstream tasks, we implement a greedy algorithm in conjunction with a submodular function to more comprehensively determine the order of the modalities. Detailed explanations and algorithms for these methods are provided in the appendix J (page 26-27), as also mentioned in the 'Response to all reviewers' comments above.
>
> Based on the algorithms mentioned above, we validated the modal selection order used in our sarcasm detection experiments. The computed sample entropies for the V/T/A modalities are 2.388/2.063/0.071, respectively. This indicates that the ranking of information richness for these modalities should be V→T→A. The outcome derived from applying the submodular function method also indicates a V→T→A sequence. The results of corresponding experiments for varying primary modalities are presented in  Section 5 (page 9) and the results are shown in Appendix F.5 (Page 23). We also present the results below. This table illustrates the effects of changing the primary modality (modal order) on weighted precision, recall, and F-score.
>
> | Metric              | V→T→A | V→A→T | T→V→A | T→A→V | A→V→T | A→T→V |
> |:---------------------|:-------:|:-------:|:-------:|:-------:|:-------:|:-------:|
> | Weighted Precision  | **75.3**  | 73.7  | 73.0  | 72.0  | 71.4  | 70.8  |
> | Weighted Recall     | **75.2** | 73.5  | 72.8  | 71.8  | 71.2  | 70.6  |
> | Weighted F-Score    | **75.2**  | 73.4  | 72.9  | 71.7  | 71.5  | 70.6  |
>
> Table 1. Varying Orders of Modalities.
>
> Table 1 showcases the impact of changing the order of modalities on the weighted Precision, Recall, and F-Score across both sarcastic and non-sarcastic classes, averaged across five folds. Underlined values highlight the best results for each metric. The modalities are denoted as: $T$ for text, $A$ for audio, and $V$ for video.
> The results show that the order of modalities, particularly the designation of the primary one, considerably influences the performance of the model. The results also indicate that the order of modalities should be V→T→A, which is consistent with the conclusions drawn from our algorithm.

---

> > ### Author Response · Authors · 2023-11-21
> > **Continue 1**
> >
> > **Comparison on recent related works** Thank you for your suggestions on including more recent state-out-of-art works. Taking into account the suggestions from you and other reviewers, we have included UniMSE [2], MIB [4] and BBFN [5] for CMU-MOSI dataset and DynMM [1] for CMU-MOSEI dataset. The comparative results for the CMU-MOSI dataset have been updated in the revised version in Table 2 (page 8) and we also provide it as follows:
> >
> > | **Task Metric** | **BA↑** | **F1↑** | **MAE↓** | **Corr↑** |
> > |-----------------|---------|---------|-----------|-----------|
> > | UniMSE [5] | 85.9 | 85.8 | 0.691 | 0.809 |
> > | MIB [7] | 85.3 | 85.3 | 0.711 | 0.798 |
> > | BBFN [8] | 84.3 | 84.3 | 0.776 | 0.755 |
> > | **ITHP** | **88.7** | **88.6** | **0.643** | **0.852** |
> > Table 2. Model Performance on CMU-MOSI dataset.
> >
> > We also compare the performance of our ITHP model with DynMM on the CMU-MOSEI dataset. DynMM achieved an accuracy of 79.8% and a Mean Absolute Error (MAE) of 0.60. In contrast, our model achieved an accuracy of 87.3% and an MAE of 0.56, surpassing DynMM by over 9.4% in accuracy and exhibiting a 6.7% lower MAE. These results further underscore the superiority of our model in terms of accuracy. We have also discussed all these models in our related work in Appendix B (page 14) as follows:
> >
> > “MIB focuses on isolating essential data features, filtering out the extraneous for purer multimodal fusion. BBFN reimagines the fusion process by merging and differentiating between modal inputs. The current SOTA model presents novel insights, UniMSE innovatively integrates multimodal sentiment analysis and emotion recognition in conversations into a unified framework, enhancing prediction accuracy by leveraging the synergies and complementary aspects of sentiments and emotions. The SPECTRA [3] is pioneering in spoken dialog understanding by pre-training on speech and text, featuring a unique temporal position prediction task for precise speech-text alignment. The DynMM creates the possibility of reducing computing resource consumption. By introducing dynamic multi-modal fusion, which fuses input data from multiple modalities adaptively, leading to reduced computation, improved representation power, and enhanced robustness. ”
> >
> > **Regarding the adjustment of model parameters across different tasks:** Thank you for raising the question about whether our model requires excessive adjustments of hyperparameters based on the task. Our answer is that such extensive adjustments are not required. In our experiments, we have tested different values of $\beta$ and $\gamma$. We found that for different experiments, these two parameters do not require much adjustment. Referring to Figure 4 (page 8), for instance, when both $\beta$ and $\gamma$ values are all smaller than 4, the model cannot retain the relevant information from the secondary modalities while losing information from the input modality with a low Lagrange multiplier, leading to worse performance. However, the results remain almost unchanged when beta and gamma are chosen within a wide range when $\beta>=8$. This implies that we have a substantial margin for parameter selection, eliminating the need for excessive adjustments. Moreover, in the experiments with the CMU-MOSI/CMU-MOSEI datasets, we keep the hyper parameters the same with ($\beta=8$ and $\gamma=32$) without tuning these hyper parameters, demonstrating the robustness of our parameter choices.
> >
> > [1] Xue Z, Marculescu R. Dynamic multimodal fusion. In Proceedings of the IEEE/CVF Conference on Computer Vision and Pattern Recognition 2023 (pp. 2574-2583).
> >
> > [2] G. Hu, et al., UniMSE: Towards Unified Multimodal Sentiment Analysis and Emotion Recognition, EMNLP 2022
> >
> > [3] T. Yu, et al., Speech-Text Dialog Pre-training for Spoken Dialog Understanding with Explicit Cross-Modal Alignment, ACL 2023
> >
> > [4] Mai S, Zeng Y, Hu H. Multimodal information bottleneck: Learning minimal sufficient unimodal and multimodal representations. IEEE Transactions on Multimedia. 2022.
> >
> > [5] Han, Wei, et al. "Bi-bimodal modality fusion for correlation-controlled multimodal sentiment analysis." Proceedings of the 2021 International Conference on Multimodal Interaction. 2021.

---

> > > ### Comment · Reviewer_4bfy · 2023-11-23
> > >
> > > Thanks to the author for detailed response.  My concerns on the order of modalities have been addressed.  My concern is also on the sequential pipeline, as the final goal is fusing all the information together, is it a better way compared to fusing all the modalities once together? When introducing the information bottleneck to percept the first two modalities, the third modality is not considered, will some complementary information to the third modality be neglect? As the sequential pipeline is one of the most important points of the paper, I would suggest a theoretical proof and experiments to demonstrate the advantage of the sequential pipeline.
> > >
> > > Currently, I keep my score.

---

> ### Author Response · Authors · 2023-11-23
> **Thank you for your response.**
>
> Thank you for your prompt response and for acknowledging the resolution of the order of modalities concern. Here we provide the point-wise response to your remaining concerns:
>
> 1.	Regarding your concern about the sequential pipeline versus fusion of all modalities once together, we wish to highlight that, while other state-of-the-art baseline models adopt strategies to fuse all modalities at once, the experiments have demonstrated the superiority of our hierarchical approach (refer to Table 1,2 and 3 in the main body of the article, Table 4 in Appendix E, Table 5 in Appendix F.1 and Table 6 in Appendix F.3).
>
> 2.	Your point about the potential loss of complementary information in the third modality is well-taken. Our method, however, ensures that no modality is neglected. Specifically, the information bottleneck is applied not only between the state $B_0$ and the second modality $X_1$ but also between the state $B_1$ and the third modality $X_2$. The entire process is data-driven, and the final information flow considers all modalities, representing the optimal solution. This guarantees that all modalities are considered, utilizing the information of the third modality (please refer to Appendix C (Pages 14-17) for the detailed design of our model).
>
> 3.	Our sequential pipeline design is not arbitrary but is grounded in neuroscience principles. We believe that the hierarchical design allows for a more natural and robust integration of modalities, which is evidenced in our experiment results compared with the baseline fusion models. We also provide the detailed explanation and theoretical basis for designing our hierarchical model based on the information bottleneck in Appendix C (Pages 14-17).
>
> We hope this response has addressed your concerns, and we sincerely appreciate your feedback, which is instrumental in enhancing the clarity of our work. We genuinely value your contributions and assure you that any further concerns will be carefully considered.

---

> > ### Comment · Reviewer_4bfy · 2023-11-23
> >
> > Thanks for the further response.  I understood that the total loss considers the third modality, and the hyper-parameters beta and gamma are introduced, that is why I asked the sensitivity to those introduced hyper-parameters. Since the proposed hierarchical design can introduce more human prior to control the information fusion process. I would suggest the authors can provide more discussion on this from theoretical perspective, rather than providing the indirect evidence of experimental results or bio-inspiration. It will make this paper stronger.  Even though my concerns are not fully addressed, I would like to increase my score to 6, since the paper has made good contribution.

---

> > > ### Author Response · Authors · 2023-11-23
> > > **Thank you for raising the score!**
> > >
> > > Thank you for carefully reviewing our response and recognizing our work as making a good contribution. We are heartened by your decision to raise your score, and we will continue to improve the work according to your insightful comments, especially through an expanded theoretical discussion on hierarchical fusion, a promising avenue for future research. We deeply value your input and are committed to diligently integrating any additional suggestions.

---

### Official Review · Reviewer_uxTC · 2023-11-02

**Soundness:** 3 good
**Presentation:** 2 fair
**Contribution:** 2 fair
**Rating:** 5
**Confidence:** 3

**Summary:**

The authors develop a new Information-Theoretic Hierarchical Perception (ITHP) model that designates a prime modality and regards the remaining modalities as detectors in the information pathway, serving to distill the flow of information. The primary modality yields the highest degree of information extraction, with subsequent modalities contributing information in a sequentially ordered manner.  To address the challenge of high dimensionality of multimodal data, they construct "Information bottlenecks." These bottlenecks function as compressed latent representations of the data, with each bottleneck responsible for compressing a single modality while retaining  the relevant information of other modalities.

The method shows impressive performance on the CMU-MOSI dataset surpassing human-level performance for sentiment analysis.

**Strengths:**

The fundamental premise of the research paper is interesting. It employs the concept of hierarchical information flow in a multi-modal context which was shown to be useful for downstream tasks like sentiment analysis and sarcasm detection.

The paper is well written and easy to understand.

**Weaknesses:**

Comparison with other methods in Table 1 for sarcasm detection is insufficient. The authors need to provide comparison results on either additional datasets [3] or other existing methods in literature [4].

[3] Cai Y, Cai H, Wan X. Multi-modal sarcasm detection in twitter with hierarchical fusion model. InProceedings of the 57th annual meeting of the association for computational linguistics 2019 Jul (pp. 2506-2515).

[4] Wen C, Jia G, Yang J. DIP: Dual Incongruity Perceiving Network for Sarcasm Detection. InProceedings of the IEEE/CVF Conference on Computer Vision and Pattern Recognition 2023 (pp. 2540-2550).

The paper is specific to the task sentiment analysis whereas the title and presentation of the paper refers to downstream tasks in multimodal learning as a whole. To support the claim the authors should report the method’s performance on some general tasks like visual question answering.

The literature survey needs to be updated. They are several recent methods of multimodal fusion and representation learning:

[1] Xue Z, Marculescu R. Dynamic multimodal fusion. InProceedings of the IEEE/CVF Conference on Computer Vision and Pattern Recognition 2023 (pp. 2574-2583).





Missing table 5 and 10 on page 9 which were referred to in text.

**Questions:**

The authors have established a direct quantitative association between the embedding size of modality X and its corresponding priority order for the task. However, the embedding size can be contingent upon the model's architectural characteristics. Does changing the model which provides the embedding has any effect on the priority order?

How is embedding size related to the amount of context information present in a modality?

For experiments in table 1, is there any effect of changing the audio and text priority, when all three of the modalities are present?

---

> ### Author Response · Authors · 2023-11-21
> **Response to Reviewer uxTC**
>
> Thank you for a thorough review of our paper and for recognizing the fundamental premise of our research. We also appreciate your recommendations of related works [1,2,3]. In our revised manuscript, we have expanded the discussion on related literature to appeal to a wider audience. Below, we provide the detailed point-wise response to address your concerns.
>
> **Comparative analysis of additional methods for sarcasm detection:** Thank you for sharing the related works [2,3] on sarcasm detection. Specifically, HFM [2] proposed a task-oriented hierarchical fusion model, which incorporates early fusion, representation fusion, and modality fusion techniques to enhance the comprehension and representation of each modality. We tested HFM using the MUStARD dataset, applying its modality fusion module under the same experimental conditions as our original study. The results, now included in Table 6 of Appendix F.3 (page 21) of the revised manuscript, are as follows:
>
> | Model | Precision | Recall | F-Score |
> |-------|-----------|--------|---------|
> | MSDM  | 71.9      | 71.4   | 71.5    |
> | HFM [2] | 69.4    | 68.8   | 68.9    |
> | ITHP  | **75.3**  | **75.2** | **75.2** |
>
> Table 1: Results of Sarcasm Detection on the MUStARD dataset.
>
> This table displays the weighted Precision, Recall, and F-Score for the V-T-A (video, text, audio) modality, averaged across five folds. Bold figures represent the highest scores achieved. As the results indicate, our ITHP model demonstrates a notable increase of 8.5% in precision, 9.3% in recall, and 9.1% in F-score compared to the HFM [2] model. These improvements underscore the superiority of our model in sarcasm detection.
>
> The DIP [3] model processes inputs consisting of two modalities: images and text. It computes a relation matrix of their feature vectors, which is then fed into two modules, SSC and SID, for further processing and eventual concat for prediction. This model architecture, specifically its requirement for modality correlation matrix computation and the use of two distinct branches for sentiment analysis, does not align well with our sarcasm datasets. Consequently, it is not feasible to conduct equivalent comparative experiments. Regardless, we appreciate your sharing of the related works, and we have also discussed them in our related work in Appendix B (page 14) to reach a broader audience as follows:
>
> “The latest work in the field of sarcasm detection has expanded our research horizons. The HFM [2] utilizes early fusion and representation fusion techniques to refine feature representations for each modality. It integrates information from various modalities to better utilize the available data and enhance sarcasm detection performance. The DIP [3] creatively adopts dual incongruity perception at factual and affective levels, enabling effective detection of sarcasm in multi-modal data.”
>
> **Literature survey update:** Thank you for introducing us to the exceptional work [1], it will greatly enhance our literature review. Diverging from the prevalent static fusion approaches, DynMM presents an innovative perspective by adaptively fusing multimodal data and creating data-dependent forward paths during inference. The main highlight of the DynMM model lies in its ability to significantly reduce computational resource consumption while maintaining relatively high model performance.
>
> We also compared the performance of our ITHP model with DynMM on the CMU-MOSEI dataset. DynMM achieved an accuracy of 79.8% and a Mean Absolute Error (MAE) of 0.60. In contrast, our model achieved an accuracy of 87.3% and an MAE of 0.56, surpassing DynMM by over 9.4% in accuracy and exhibiting a 6.7% lower MAE. These results further underscore the superiority of our model in terms of accuracy. We have also discuss DynMM in in Appendix B (page 14):
>
> “The DynMM [1] creates the possibility of reducing computing resource consumption. By introducing dynamic multi-modal fusion, which fuses input data from multiple modalities adaptively, leading to reduced computation, improved representation power, and enhanced robustness.”
>
> [1] Xue Z, Marculescu R. Dynamic multimodal fusion. InProceedings of the IEEE/CVF Conference on Computer Vision and Pattern Recognition 2023 (pp. 2574-2583).
>
> [2] Cai Y, Cai H, Wan X. Multi-modal sarcasm detection in twitter with hierarchical fusion model. InProceedings of the 57th annual meeting of the association for computational linguistics 2019 Jul (pp. 2506-2515).
>
> [3] Wen C, Jia G, Yang J. DIP: Dual Incongruity Perceiving Network for Sarcasm Detection. InProceedings of the IEEE/CVF Conference on Computer Vision and Pattern Recognition 2023 (pp. 2540-2550).

---

> > ### Author Response · Authors · 2023-11-21
> > **Continue 1**
> >
> > **Exploring the Scope of Multimodal Learning Beyond Sentiment Analysis:** Thank you for suggesting more experiments on other tasks beyond sentiment analysis. Recognizing the vastness of multimodal learning beyond sentiment analysis [4, 5], in our work, we chose the CMU-MOSI, and CMU-MOSEI datasets for their significant role and widespread applicability in this domain. These datasets are chosen for their extensive applicability and significance in the domain, facilitating meaningful comparisons with the current state-of-the-art models [6 - 9]. In addition, our model has the potential to be extended to other multimodal tasks, which will be explored further in the future.
> >
> > **For table 5 and 10 that referred to in the text:** The tables were placed in the Appendix. Table 5 “Model Performance on CMU-MOSI” has been placed in Appendix F.1 (page 20). Table 10 (due to the revision, this table is currently labeled as Table 11) “Varying Orders of Modalities” has been placed in the Appendix F.5 (page 23). We will indicate this in the text on page 9 for easy reference by readers.
> >
> > [4] Zadeh, Amir, et al. "Mosi: multimodal corpus of sentiment intensity and subjectivity analysis in online opinion videos." arXiv preprint arXiv:1606.06259 (2016).
> >
> > [5] Zadeh, AmirAli Bagher, et al. "Multimodal language analysis in the wild: Cmu-mosei dataset and interpretable dynamic fusion graph." Proceedings of the 56th Annual Meeting of the Association for Computational Linguistics (Volume 1: Long Papers). 2018.
> >
> > [6] UniMSE : G. Hu, et al., UniMSE: Towards Unified Multimodal Sentiment Analysis and Emotion Recognition, EMNLP 2022
> >
> > [7] Sijie Mai, Ying Zeng, and Haifeng Hu. Multimodal information bottleneck: Learning minimal sufficient unimodal and multimodal representations. IEEE Transactions on Multimedia, 2022.
> >
> > [8] Wei Han, Hui Chen, Alexander Gelbukh, Amir Zadeh, Louis-philippe Morency, and Soujanya Poria. Bibimodal modality fusion for correlation-controlled multimodal sentiment analysis. In Proceedings of the 2021 International Conference on Multimodal Interaction, pp. 6–15, 2021a.
> >
> > [9] Rahman, Wasifur, et al. "Integrating multimodal information in large pretrained transformers." Proceedings of the conference. Association for Computational Linguistics. Meeting. Vol. 2020. NIH Public Access, 2020.

---

> > > ### Author Response · Authors · 2023-11-21
> > > **Continue 2**
> > >
> > > **Embedding size and modal order:** We appreciate your insights regarding the correlation between the embedding sizes of modalities and their priority order, as discussed in our paper. It's important to note that this correlation is an initial observation and should not be viewed as an exhaustive or final conclusion. The prioritization of modalities in reality should be based on the richness of the information they provide. After carefully considering your comments along with those from other reviewers, we have developed more general approaches to define the order of modalities: (i) We employ Sample Entropy (SampEn) to assess the information richness within different modalities, which serves as the basis for ranking these modalities. (ii) For specific downstream tasks, we implement a greedy algorithm in conjunction with a submodular function to more comprehensively determine the order of the modalities. Detailed explanations and algorithms for these methods are provided in the appendix J (page 26-27), as also mentioned in the 'Response to all reviewers' comments above.
> > >
> > > Moreover, regarding your questions about changing the embedding models: Altering the embedding model would certainly modify the embedding sizes. Nonetheless, the priority order of the modalities and the richness of information they carry still need to be calculated and decided using the methods we proposed in the appendix J (page 26-27) as we mentioned above. The experiments related to determining the order of the modalities are discussed in the following answer.
> > >
> > >
> > > **Experiments with different orders of modalities:**
> > > We appreciate your question regarding the selection of the primary modality and its impact on the ITHP model's performance. In fact, we have already conducted the experiments for varying primary modalities as highlighted in Section 5 (page 9) and the results are shown in Appendix F.5 (Page 23). We also present the results below. This table illustrates the effects of changing the primary modality (modal order) on weighted precision, recall, and F-score.
> > >
> > > | Metric              | V→T→A | V→A→T | T→V→A | T→A→V | A→V→T | A→T→V |
> > > |:---------------------|:-------:|:-------:|:-------:|:-------:|:-------:|:-------:|
> > > | Weighted Precision  | **75.3**  | 73.7  | 73.0  | 72.0  | 71.4  | 70.8  |
> > > | Weighted Recall     | **75.2** | 73.5  | 72.8  | 71.8  | 71.2  | 70.6  |
> > > | Weighted F-Score    | **75.2**  | 73.4  | 72.9  | 71.7  | 71.5  | 70.6  |
> > >
> > > Table 2. Varying Orders of Modalities.
> > >
> > > The table showcases the impact of changing the order of modalities on the weighted Precision, Recall, and F-Score across both sarcastic and non-sarcastic classes, averaged across five folds. Underlined values highlight the best results for each metric. The modalities are denoted as: $T$ for text, $A$ for audio, and $V$ for video.
> > >
> > > Table 2 shows that the order of modalities, particularly the designation of the primary one, considerably influences the performance of the model. Based on the algorithms mentioned above (i.e., SampEn method and submodular function method) we proposed, we validated the modal selection order used in our experiments. The computed SampEn for the V/T/A modalities are 2.388/2.063/0.071, respectively. This indicates that the ranking of information richness for these modalities should be V→T→A. The outcome derived from applying the submodular function method also indicates a V→T→A sequence. Hence, V should be chosen as the primary modality and the order should be V→T→A. This aligns with the results obtained in our experiments, confirming the viability of our proposed detection criteria.

---

> > > > ### Comment · Reviewer_uxTC · 2023-11-23
> > > >
> > > > The authors have addressed the concerns regarding the order of modality to be considered during training. They have proposed a greedy based approach which is more general than depending on the embedding sizes. The literature and comparisons with recent baselines has been updated as well. They have reported the validation of SampEn and greedy algorithm for sarcasm detection but I think since the entire hypothesis of the paper is dependent on the order of modality the results should also have been validated for the chosen T-A-V for sentiment analysis task. It is difficult to make a concrete comment with results on a single dataset (Mustard) which has kind of an obvious order (V-T-A).
> > > >
> > > > I keep my score.

---

> > > > > ### Author Response · Authors · 2023-11-23
> > > > > **Updating Validated Results for Sentiment Analysis Task Soon**
> > > > >
> > > > > Thank you for recognizing our efforts in addressing the concerns about the order of modality and for the updates to the baselines. We appreciate your suggestion to validate the ranking algorithm for the chosen T-A-V (Text-Audio-Visual) for the sentiment analysis task. We are currently conducting these experiments and will provide an update soon.

---

> > > > > > ### Author Response · Authors · 2023-11-23
> > > > > > **Further Discussion**
> > > > > >
> > > > > > Thank you for your insightful suggestion. Following your advice, we have applied our proposed modality ranking algorithms to the sentiment analysis task. (i) In our experiments on the MOSI dataset, both algorithms indicated a T->A->V (Text->Audio->Video) sequence. The Sample Entropy values obtained were 0.7856 for Text, 0.02978 for Audio, and 0.01418 for Video. (ii) On the MOSEI dataset, both algorithms again suggested a T->A->V order, with the respective Sample Entropy values 1.0663 for Text, 0.06213 for Audio, and 0.02479 for Video. We hope this addresses your current concerns. We genuinely value your contributions and assure you that any further suggestions will be carefully considered and incorporated.

---

### Official Review · Reviewer_Qqaa · 2023-11-07

**Soundness:** 3 good
**Presentation:** 2 fair
**Contribution:** 3 good
**Rating:** 6
**Confidence:** 3

**Summary:**

- This paper introduces the Information-theoretic Hierarchical Perception (ITHP) model, which aims to integrate and process information from multiple modalities effectively. The authors show inspiring motivation corresponding neuroscience research to a model that designates a prime modality and utilizes the concept of the information bottleneck to construct compact and informative latent states, which enable to balance between preserving relevant information and reducing noise in the latent states.

- This paper tries to justify the validity of design philosophy and the effectiveness of the proposed models experimentally with the performance comparison of two tasks such as (1) sarcasm detection from videos and (2) multimodal sentiment analysis. They seem to show better scores in both tasks. (note that seems not to pursue state-of-the-art performances).

**Strengths:**

- This paper presents interesting connections between neuroscience and proposed models. It provides good motivations for designing the proposed models.﻿ This integration of neuroscience principles into the design of the models adds a novel perspective to the field of multimodal learning.

- The authors demonstrate the effectiveness with competitive performances of the proposed models in the two tasks.

**Weaknesses:**

- Even though this paper shows interesting modeling design and experimental results, there is a gap between state-of-the-art methods such as UniMSE and SPECTRA, for example, with respect to performances. What kinds of criteria to choose the model configuration and comparative methods in the paper?

- This paper seems not to present enough information on experimental setting and implementation to achieve reproducibility.

- It lacks justification for the component choices. Which points should be clarified through experiments? Why BERT and DeBERTa are utilized as language models?


* References

  - UniMSE : G. Hu, et al., UniMSE: Towards Unified Multimodal Sentiment Analysis and Emotion Recognition, EMNLP 2022

  - SPECTRA : T. Yu, et al., Speech-Text Dialog Pre-training for Spoken Dialog Understanding with Explicit Cross-Modal Alignment, ACL 2023


----------------------------------------------------------------------------
Post-rebuttal
----------------------------------------------------------------------------
I read the rebuttal and other reviewers' opinions.
At first, the rebuttal addresses most of my major concerns and provides convincing information to resolve mine.
I'd like to raise my score from 5 to 6.
Thank you for your good contributions to our community!

**Questions:**

See the Weaknesses above and answer, please.

---

> ### Author Response · Authors · 2023-11-21
> **Response to Reviewer Qqaa**
>
> We sincerely thank you for highlighting the good motivations behind our models and recognizing how their integration of neuroscience principles adds a novel perspective to multimodal learning. We also thank you for sharing these two interesting related works [1,2], which we have discussed in the revised version to reach a broader audience. Additionally, we provide detailed answers to address your questions below.
>
>
> **Regarding the comparison between our method, ITHP, against SOTA methods such as UniMSE and SPECTRA:**  Thank you for sharing these related works [1,2], we have first added them into the references and discussed them in the related work section in Appendix B (page 14) as follows:
> “The current state-of-the-art model presents novel insights, UniMSE [1]  innovatively integrates multimodal sentiment analysis and emotion recognition in conversations into a unified framework, enhancing prediction accuracy by leveraging the synergies and complementary aspects of sentiments and emotions. SPECTRA [2] is pioneering in spoken dialog understanding by pre-training on speech and text, featuring a unique temporal position prediction task for precise speech-text alignment .”
>
> We have also expanded our baselines to include more recent works such as UniMSE [1], MIB [3] and BBFN [4] for a more comprehensive comparison, as detailed in the updated Table 2 (page 8) in the paper. The results show that our model continues to surpass the baselines.
>
> Regarding the comparison with the models you mentioned, particularly UniMSE [1] and SPECTRA [2]: First, regarding performance, we outperform UniMSE on CMU-MOSEI and CMU-MOSI, two common evaluation benchmarks shared by both of the works, by around 1% and 3% points in terms of F1 score. Meanwhile, SPECTRA is surpassed by 1% in MOSI datasets, demonstrating our model’s superiority. Here we would like to point out that our data preprocessing is based on the previous paper MAG [5], which only provides off-the-shelf representations of the images and audio. However, both UniMSE and SPECTRA, adopted a different pipeline (e.g., in UniMSE, they use more powerful tools such as effecientNet to extract features from the video, coupled with additional A/V-sLSTM to further distill the multimodal signals). In this sense, the comparison with Self-MM, MMIM, and MAG are considered as more fair baselines given the same pre-processing pipelines, where ITHP has dominated in performance against all of the above.
> Second, our ITHP is an information-theoretic with guaranteed proof of aggregating the most useful information across different modalities. Previous works, such as UniMSE and SPECTRA, either *empirically* instead of *theoretically* adopts contrastive loss to force the latent space to be as distinctive as possible, or *empirically* aligns sequential audio-text features which are not generalizable to images.
>
> [1] UniMSE : G. Hu, et al., UniMSE: Towards Unified Multimodal Sentiment Analysis and Emotion Recognition, EMNLP 2022
>
> [2] SPECTRA : T. Yu, et al., Speech-Text Dialog Pre-training for Spoken Dialog Understanding with Explicit Cross-Modal Alignment, ACL 2023
>
> [3] MIB: Mai S, Zeng Y, Hu H. Multimodal information bottleneck: Learning minimal sufficient unimodal and multimodal representations. IEEE Transactions on Multimedia. 2022.
>
> [4] Han, Wei, et al. "Bi-bimodal modality fusion for correlation-controlled multimodal sentiment analysis." Proceedings of the 2021 International Conference on Multimodal Interaction. 2021.
>
> [5] Rahman, Wasifur, et al. "Integrating multimodal information in large pretrained transformers." Proceedings of the conference. Association for Computational Linguistics. Meeting. Vol. 2020. NIH Public Access, 2020.

---

> > ### Author Response · Authors · 2023-11-21
> > **Continue 1**
> >
> > **Clarity on reproducibility of experimental setup:** We understand your concern for details of our experimental setup, which is critical for reproducibility. It should be noted that we have indeed provided the source code link for our model and experiments in Appendix A "CODE AVAILABILITY" (page 14). At the same time, we have listed the selection of model parameters in the experimental part in detail in the Appendix E "MODEL ARCHITECTURE AND HYPERPARAMETERS" (page 17-18). We also summarize the description here for convenience:
> >
> > “Here we provide a detailed description of our model architecture and the hyperparameter selection in different tasks for reproducibility purposes. For the ITHP, to get the latent state's distribution of $B_0$, we directly use two fully connected linear layers to build the encoder, where the first layer extracts the feature vectors of input $X_0$ and the second layer outputs parameters for a latent Gaussian distribution - a mean vector and the logarithm of the variance vector. By sampling from this distribution, we obtain a latent state that represents the input state further used to (1) feed into the multilayer perceptron (MLP) layer to predict $X_1$; (2) feed into the next hierarchical layer for another modality information. Note that this is a serial structure, and the remaining layers share the same design. The predictor is chosen specifically by the downstream tasks. For the task of sarcasm detection, we elect to use an MLP with two layers as the predictor. For the sentiment analysis task, the model ITHP-DeBERTa is trained in an end-to-end manner. The raw features of acoustic and visual modalities are pre-extracted from CMU-MOSI [6]. The textual features are extracted by pre-trained DeBERTa [7, 8]. The textual features serve as input to our ITHP module while the pre-extracted acoustic and visual act as the detectors. For the task of sarcasm detection, unless otherwise specified, we set the  hyperparameters as follows: $\beta=32, \gamma=8, \lambda=1$. We perform a 5-fold cross-validation, and for each experiment, we train the ITHP model for 200 epochs using an Adam optimizer with a learning rate of $10^{-3}$. For the task of sentiment analysis, unless otherwise specified, we set the  hyperparameters as follows: $\beta=8, \gamma=32, \lambda=1$. We run each experiment for 40 epochs using the Adam optimizer with a learning rate of $10^{-5}$. We repeat each experiment 5 times to calculate the mean value of the metrics.”
> >
> > We are committed to making our code and datasets public to ensure that our work can be replicated and verified by others in the field.
> >
> >
> > **Rationale for component selection:** Thank you for pointing out your concerns on component choice. Your point about the rationale for our component selection, specifically the rationale for using BERT and DeBERTa, is easily accepted. Though our method ITHP is a **model-agnostic** method, we choose BERT (RoBERTa) and DeBERTa as they are the **go-to** method for text encoding and also have some benefits such as lightweight (hundred-million-scale parameter size). They are widely used in most of the text-encoding/natural language understanding related works ranging from question answering, sentiment analysis, commonsense reasoning , and so on. In fact, the adoption of DeBERTa instead of merely RoBERTa is a demonstration of generalizability across different methods for text encoding. We leave the adaptations to other text encoders, such as T5, Electra, and even more recent LLMs such as Llama, GPT3, as a future work to better understand the generalizability of our work.
> >
> > [6] Zadeh, Amir, et al. "Mosi: multimodal corpus of sentiment intensity and subjectivity analysis in online opinion videos." arXiv preprint arXiv:1606.06259 (2016).
> >
> > [7] He, Pengcheng, et al. "Deberta: Decoding-enhanced bert with disentangled attention." arXiv preprint arXiv:2006.03654 (2020).
> >
> > [8] He, Pengcheng, Jianfeng Gao, and Weizhu Chen. "Debertav3: Improving deberta using electra-style pre-training with gradient-disentangled embedding sharing." arXiv preprint arXiv:2111.09543 (2021).

---

### Official Review · Reviewer_mWBz · 2023-11-08

**Soundness:** 3 good
**Presentation:** 3 good
**Contribution:** 3 good
**Rating:** 8
**Confidence:** 3

**Summary:**

This paper proposes ITHP, a brain-inspired hierarchical information perception model that fuses information from different modalities. ITHP employs the information bottleneck framework to extract relevant information from different modalities in a hierarchical/sequential manner. Extensive experiments on MUStARD, CMU-MOSI, and CMU-MOSEI demonstrate the effectiveness of ITHP.

**Strengths:**

- the paper is clear and well-organized.
- the proposed ITHP is intuitive and well-grounded in information theory and provides a novel view of information fusing in multi-modal learning.
- extensive experiments demonstrate the effectiveness of ITHP.

**Weaknesses:**

- ITHP extracts commonly encoded information from different modalities; however, would this procedure be called "fusion" appropriately?
    - suppose the primary modality is $M_p$ and a secondary modality be $M_1$. ITHP cannot capture information encoded in $M_1$ but not $M_p$. In this sense, ITHP would be an information distillation rather than a fusion method.
    - when talking about "fusion," one expects to integrate information of **different** types.
    - in addition, when both modalities contain similar information, if the one in $M_1$ is noisy, would ITHP be affected? Can ITHP correctly identify the noise in $M_1$ and mainly use the information from $M_p$ instead?
    - as the main idea of ITHP is to distill relevant information from different modalities, the starting point would be important. However, the experiments do not show results with different primary modalities.

**Questions:**

- how do you choose the primary modality? And how would different choices of primary modality affect the performance of ITHP?
- how would ITHP perform in the presence of noise in some of the modalities (the primary or the secondary ones)?
- how can ITHP accommodate tasks that require combining complementary information from different modalities?

---

> ### Author Response · Authors · 2023-11-21
> **Response to Reviewer mWBz**
>
> Thank you for your meticulous review and valuable insights. We sincerely appreciate your recognition of the ITHP model’s intuitiveness and its strong grounding in information theory. In the following, we offer our detailed responses and clarifications to address the concerns and queries you have highlighted.
>
> **The usage of "fusion" in our method:** Thank you for highlighting the usage of the term “fusion” in our study. We are grateful for your detailed attention to this aspect, and we have thoroughly reviewed our manuscript to ensure that this term is used accurately and appropriately. Fusion models primarily aim to integrate information from various modalities into a unified representation [1,2]. In terms of this objective, our work can certainly be viewed as a fusion model constructing a compact representation by utilizing information from various modalities. In terms of technique, we indeed adopt a distinct hierarchical approach compared to traditional fusion models, which stands as a significant innovation in our work. Although the secondary modalities serve as detectors in our model, it is worthy noting that, within the deep learning architecture, our model has certain capability to indirectly learn the complementary information from secondary modalities. As explained in Appendix H (page 24):” by harnessing the capabilities of our deep learning framework, our model has a certain ability to learn the complementary information from secondary modalities. To achieve a harmonious representation, we’ve incorporated specific terms in the loss function. Specifically, taking the 3-modalities learning problem as an example, term $L_{1} =−\beta\cdot I\left (B_{0};X_{1} \right) $ is tasked with supervising the knowledge transfer from $X_{1}$ to $B_{0}$, while term $L_{2} =−\gamma\cdot I\left (B_{1};X_{2} \right) $ is responsible for supervising the knowledge dissemination from $X_{2}$ to $B_{1}$. Through this framework, the ITHP model is primed to discern and incorporate complementary information from non-input modalities.” In summary, we prefer to use the term 'fusion' in the context of our mode, and we will make it clear in our revised version.
>
> [1] Gao, Jing, et al. "A survey on deep learning for multimodal data fusion." Neural Computation 32.5 (2020): 829-864.
>
> [2] Atrey, Pradeep K., et al. "Multimodal fusion for multimedia analysis: a survey." Multimedia systems 16 (2010): 345-379.

---

> ### Author Response · Authors · 2023-11-21
> **Continue 1**
>
> **Model performance in the presence of noise:** Thanks for the insightful suggestions on adding noise to the modality $M_{p}$ and modality $M_{1}$. We believe these experiments will bring valuable insights to further elucidate our model. Theoretically, our model filters redundant information within its information-bottleneck hierarchical structure, and we anticipate it will have advantages over traditional fusion models in terms of reducing the impact of noise. According to your suggestions, we conducted the noise addition experiments as follows:
>
> In the sarcasm detection dataset, we introduced noise to the normalized data of both the primary and secondary modalities separately to test the model's resistance to noise interference.  We apply a series of Gaussian noises with a mean of 0 and a varying standard deviation ranging from 0.05 to 0.3, increasing in increments of 0.05. For each noise level, we repeated the experiment 50 times and calculated the average result. We have included these experiments in the revised version (Please refer to Appendix F.6, page 23-24), and the results are presented below:
>
>
> | $\sigma$ | -     | 0.05  | 0.1   | 0.15  | 0.2   | 0.25  | 0.3   |
> |----------------------|-------|-------|-------|-------|-------|-------|-------|
> | **MSDM Precision** | 0.719 | 0.711 (-1.11%) | 0.709 (-1.39%) | 0.707 (-1.67%) | 0.704 (-2.09%) | 0.700 (-2.64%) | 0.688 (-4.31%) |
> | **MSDM Recall** | 0.714 | 0.705 (-1.26%) | 0.702 (-1.68%) | 0.700 (-1.96%) | 0.695 (-2.66%) | 0.691 (-3.22%) | 0.679 (-4.90%) |
> | **ITHP Precision** | 0.753 | 0.751 (-0.27%) | 0.748 (-0.66%) | 0.747 (-0.80%) | 0.746 (-0.93%) | 0.743 (-1.33%) | 0.740 (-1.73%) |
> | **ITHP Recall** | 0.752 | 0.750 (-0.27%) | 0.745 (-0.93%) | 0.743 (-1.20%) | 0.742 (-1.33%) | 0.740 (-1.60%) | 0.736 (-2.13%) |
>
> Table 1. Evaluating Gaussian Noise Impact on Sarcasm Detection in Text Modality.
>
> Table 1 presents the weighted precision and recall for sarcasm detection on the MUStARD dataset, with an analysis of Gaussian noise addition to the text ($M_{1}$) modality. Results are averaged over five cross-validation folds and 50 trials for each level of noise intensity. Each column indicates the standard deviation (represented by $\sigma$) of the introduced Gaussian noise with a mean of 0. Changes in performance metrics relative to the noise-free baseline are expressed as percentages
>
>
>
> | $\sigma$ | -      | 0.05     | 0.1      | 0.15     | 0.2      | 0.25      | 0.3       |
> |----------------|--------------|------------|-----------|------------|------------|------------|-------------|
> | **MSDM Precision**| 0.719         | 0.709 (-1.39%)    | 0.705 (-1.95%)    | 0.702 (-2.36%)    | 0.698 (-2.92%)    | 0.692 (-3.76%)    | 0.683 (-5.01%)    |
> | **MSDM Recall**   | 0.714         | 0.703 (-1.54%)    | 0.700 (-1.96%)    | 0.694 (-2.80%)    | 0.689 (-3.50%)    | 0.678 (-5.04%)    | 0.668 (-6.44%)    |
> | **ITHP Precision**    | 0.753         | 0.748 (-0.66%)    | 0.745 (-1.06%)    | 0.743 (-1.33%)    | 0.742 (-1.46%)    | 0.739 (-1.86%)    | 0.735 (-2.39%)    |
> | **ITHP Recall**       | 0.752         | 0.745 (-0.93%)    | 0.741 (-1.46%)    | 0.740 (-1.60%)    | 0.738 (-1.86%)    | 0.733 (-2.53%)    | 0.730 (-2.93%)    |
>
> Table 2. Evaluating Gaussian Noise Impact on Sarcasm Detection in Video Modality.
>
> This table presents the weighted precision and recall for sarcasm detection on the MUStARD dataset, with an analysis of Gaussian noise addition to the video ($M_{p}$) modality. Results are averaged over five cross-validation folds and 50 trials for each level of noise intensity. Each column indicates the standard deviation (represented by $\sigma$) of the introduced Gaussian noise with a mean of 0. Changes in performance metrics relative to the noise-free baseline are expressed as percentages
>
> The results shown in Table 1 and Table 2 demonstrate that, compared to the baseline model, our model exhibits greater robustness to noise interference. The impact of adding noise to the primary modality is more significant than the effect of noise on the secondary modality.

---

> ### Author Response · Authors · 2023-11-21
> **Continue 2**
>
> **Experiments with different primary modalities:**
> We appreciate your question regarding the selection of the primary modality and its impact on the ITHP model's performance. In fact, we have already conducted the experiments for varying primary modalities as highlighted in  Section 5 (page 9) and the results are shown in Appendix F.5 (Page 23). We also present the results below. This table illustrates the effects of changing the primary modality (modal order) on weighted precision, recall, and F-score.
>
> | Metric              | V→T→A | V→A→T | T→V→A | T→A→V | A→V→T | A→T→V |
> |:---------------------|:-------:|:-------:|:-------:|:-------:|:-------:|:-------:|
> | Weighted Precision  | **75.3**  | 73.7  | 73.0  | 72.0  | 71.4  | 70.8  |
> | Weighted Recall     | **75.2**  | 73.5  | 72.8  | 71.8  | 71.2  | 70.6  |
> | Weighted F-Score    | **75.2**  | 73.4  | 72.9  | 71.7  | 71.5  | 70.6  |
>
> Table 3. Varying Orders of Modalities.
>
> The table showcases the impact of changing the order of modalities on the weighted Precision, Recall, and F-Score across both sarcastic and non-sarcastic classes, averaged across five folds. Underlined values highlight the best results for each metric. The modalities are denoted as: $T$ for text, $A$ for audio, and $V$ for video.
>
> The results show that the order of modalities, particularly the designation of the primary one, considerably influences the performance of the model. In many cases, we can rely on prior knowledge to guide the selection of the primary modality and its sequence. However, when such knowledge is lacking, effective methods are essential for determining these aspects. In response to the feedback from reviewers, we have developed practical algorithms to define the primary modality and the order of modalities: (i) We employ Adapting Sample Entropy (SampEn) to assess the information richness within different modalities, which serves as the basis for ranking these modalities. (ii) For specific downstream tasks, we implement a greedy algorithm in conjunction with a submodular function to more comprehensively determine the order of the modalities. Detailed explanations and algorithms for these methods are provided in the appendix J (page 26-27), as also mentioned in the 'Response to all reviewers' comments above.
>
> Based on the methods for modalities ranking we proposed, we validated the modal selection order used in our experiments. For the first method, the computed SampEn for the V/T/A modalities are 2.388/2.063/0.071, respectively. This indicates that the ranking of information richness for these modalities should be V→T→A. Additionally, by applying the algorithm (ii), we obtained the same modality order of V→T→A. Hence, we chose V as our primary modality. This aligns with the results obtained in our experiments, confirming the viability of our proposed detection criteria.
>
> **Learning complementary information:** As we mentioned in the above response (**The usage of "fusion" in our method**), our ITHP model, built within the deep learning architecture has a certain ability to learn the complementary information from secondary modalities. Please refer to Appendix H (page 24) for details.

---

### Author Response · Authors · 2023-11-21
**Response to all reviewers**

We sincerely thank all the reviewers for their thorough review and insightful feedback on our manuscript. We are encouraged by the reviewers’ recognition of the novelty in our work. The reviewers’ suggestions have also led us to improve the clarity of our paper further. Here, we provide a common response to all the reviewers, and we respond to each reviewer's comments in detail accordingly behind. In the revised version of our paper, all changes have been marked in blue.
1. **Additional experiments and results.** Following the reviewers' suggestions, we have added the following experimental results to enhance our paper.
**- Noisy impact on modalities.** (**Reviewer mWBz**)
We have added Gaussian noise separately to the primary modality (Video Modality) and secondary modality (Text Modality) and validated the performance of the model on the MUStARD dataset.
**- New baselines.**（**Reviewers Qqaa and uxTC**）
We have incorporated the HFM [1] model for the MUStARD dataset and UniMSE [2], MIB [3], and BBFN [4] models for the CMU-MOSI model to achieve a more comprehensive comparison.
**- Latency time.** (**Reviewer JgMF**)
We have conducted experiments to assess the inference latency of our ITHP model with hierarchical structures.
2. **Additional references.** （**Reviewers Qqaa, uxTC and 4bfy**）
In line with the reviewers' valuable feedback, we have expanded the related work section in our paper with more citations to provide a more comprehensive perspective and reach for broader audiences.
3. **Standard criteria for determining order of modalities.** (**Reviewers mWBz, Qqaa, uxTC and 4bfy**)
Based on the suggestions from the reviewers, we believe that practical guidance for determining the order of modalities is crucial in our work. We use the richness of information as the criterion for ranking multiple modalities, which is particularly important for determining the order of modalities when prior knowledge is unreachable. To this end, we propose the following two approaches: (i) We employ Adapting Sample Entropy (SampEn) [5,6,7] to assess the information richness within different modalities, which serves as the basis for ranking these modalities. (ii) For specific downstream tasks, we implement a greedy algorithm in conjunction with a submodular function [8] to more comprehensively determine the order of the modalities. Detailed explanations and algorithms for these methods are provided in the appendix J (page 26), and we also provide the explanations and pseudocode for these two algorithms below.

(i) **Adapting sample entropy (SampEn) for datasets**

The SampEn of a dataset is calculated as follows:
- Choose an embedding dimension $ m = 2 $ and a tolerance $ r = 0.2 \times \text{standard deviation of the data} $.
-  For each sample in the dataset, form vectors $ U^m_i $ for $ i = 1, \ldots, D-m $ where $ D $ is the number of features.
-  Define $ B^m(r) $ as the count of times when two vectors $ U^m_i $ and $ U^m_{i+1} $ within the same sample are within $ r $ of each other.
-  Define $A^m(r)$ similarly for $m+1$.
-  SampEn is given by: $text{SampEn}(m, r, N) = -\ln\left(\frac{A^m(r)}{B^m(r)}\right) $

---
$$
\\begin{align*}
& \\textbf{Algorithm 1: Ranking Modalities Based on SampEn} \\\\
& \\text{function} \\ \\textbf{SampleEntropy}(data, m, r) \\\\
& \\quad N \\gets \\text{number of samples in } data \\\\
& \\quad B^m \\gets 0 \\\\
& \\quad A^m \\gets 0 \\\\
& \\quad \\text{for each sample in data} \\\\
& \\quad \\quad \\text{for } i \\gets 1 \\text{ to number of features } - m \\\\
& \\quad \\quad \\quad \\text{if norm(sample}[i:i+m] - \\text{sample}[i+1:i+m+1]) < r \\\\
& \\quad \\quad \\quad \\quad B^m \\gets B^m + 1 \\\\
& \\quad \\quad \\quad \\quad \\text{if } i < \\text{number of features } - m - 1 \\text{ and norm(sample}[i:i+m+1] - \\\\
& \\quad \\quad \\quad \\quad \\text{sample}[i+1:i+m+2]) < r \\\\
& \\quad \\quad \\quad \\quad \\quad A^m \\gets A^m + 1 \\\\
& \\quad \\quad \\quad \\quad \\text{end if} \\\\
& \\quad \\quad \\quad \\text{end if} \\\\
& \\quad \\quad \\text{end for} \\\\
& \\quad \\text{end for} \\\\
& \\quad \\text{return } -\\log(A^m / B^m) \\quad \\textit{// SampEn value} \\\\
& \\text{end function} \\\\
\\end{align*}
$$

---

$$
\\begin{align*}
& \\text{function} \\ \\textbf{RankModalities}(modalities, m, r) \\\\
& \\quad ModalityList \\gets [ ] \\\\
& \\quad \\text{for each } modality \\text{ in modalities} \\\\
& \\quad \\quad entropy \\gets \\textbf{SampleEntropy}(modality, m, r) \\\\
& \\quad \\quad \\text{append } (modality, entropy) \\text{ to ModalityList} \\\\
& \\quad \\text{end for} \\\\
& \\quad \\text{sort ModalityList based on entropy in descending order} \\\\
& \\quad \\text{return ModalityList} \\quad \\textit{// List of modalities ranked by entropy} \\\\
& \\text{end function}
\\end{align*}
$$

---

---

> ### Author Response · Authors · 2023-11-21
> **Continue 1**
>
> (ii)  **Ranking modalities using submodular functions**
>
> Given a set of $ N $ different modalities $ M = \{m_1, m_2, \ldots, m_N\} $, the goal is to find a subset of modalities $ S \subseteq M $ that maximizes the predictive performance in a given downstream task.
> Submodular Function Definition: Let $ f: 2^M \rightarrow \mathbb{R} $ be a submodular function that maps a set of modalities to a real number, indicating their performance in a predictive task. For any $ A \subseteq B \subseteq M $ and $ x \notin B $, the function $ f $ satisfies:
>
> $$ f(A \cup \{x\}) - f(A) \geq f(B \cup \{x\}) - f(B) $$
>
> Greedy Algorithm: The greedy algorithm starts with an empty set, and iteratively adds a modality that maximizes the function $ f(S) $:
>
> $$ S_{i+1} = S_i \cup \{\arg\max_{x \notin S_i} f(S_i \cup \{x\})\} $$
>
> ---
> $$
> \\begin{align*}
> & \\textbf{Algorithm 2: Greedy Algorithm for Ranking Modalities} \\\\
> & \\textbf{function} \\ \\text{GreedySubmodularSort}(modalities, evaluatePerformance) \\\\
> & S \\gets \\text{empty set} \\\\
> & ModalityList \\gets \\text{empty list} \\\\
> & \\textbf{while} \\ |S| < | modalities | \\ \\textbf{do} \\\\
> & \\quad bestImprovement \\gets -\\infty \\\\
> & \\quad bestModality \\gets \\text{null} \\\\
> & \\quad \\textbf{for each} \\ modality \\ \\text{ in modalities} - S \\ \\textbf{do} \\\\
> & \\qquad newPerformance \\gets evaluatePerformance(S \\cup \\{modality\\}) \\\\
> & \\qquad improvement \\gets newPerformance - evaluatePerformance(S) \\\\
> & \\qquad \\textbf{if} \\ improvement > bestImprovement \\ \\textbf{then} \\\\
> & \\qquad \\quad bestImprovement \\gets improvement \\\\
> & \\qquad \\quad bestModality \\gets modality \\\\
> & \\qquad \\textbf{end if} \\\\
> & \\quad \\textbf{end for} \\\\
> & \\quad \\textbf{if} \\ bestModality \\ \\text{is null} \\ \\textbf{then} \\\\
> & \\quad \\quad \\textbf{break} \\\\
> & \\quad \\textbf{end if} \\\\
> & \\quad S \\gets S \\cup \\{bestModality\\} \\\\
> & \\quad ModalityList.\\text{append}(bestModality) \\\\
> & \\textbf{end while} \\\\
> & \\textbf{return} \\ ModalityList \\\\
> & \\textbf{end function} \\\\
> \\end{align*}
> $$
>
> ---
>
> [1] Cai Y, Cai H, Wan X. Multi-modal sarcasm detection in twitter with hierarchical fusion model. InProceedings of the 57th annual meeting of the association for computational linguistics 2019 Jul (pp. 2506-2515).
>
> [2]  G. Hu, et al., UniMSE: Towards Unified Multimodal Sentiment Analysis and Emotion Recognition, EMNLP 2022
>
> [3] Mai S, Zeng Y, Hu H. Multimodal information bottleneck: Learning minimal sufficient unimodal and multimodal representations. IEEE Transactions on Multimedia. 2022.
>
> [4] Han, Wei, et al. "Bi-bimodal modality fusion for correlation-controlled multimodal sentiment analysis." Proceedings of the 2021 International Conference on Multimodal Interaction. 2021.
>
> [5] Zurek, Sebastian, et al. "On the relation between correlation dimension, approximate entropy and sample entropy parameters, and a fast algorithm for their calculation." Physica A: Statistical Mechanics and its Applications 391.24 (2012): 6601-6610.
>
> [6] Pincus, Steven M. "Approximate entropy as a measure of system complexity." Proceedings of the National Academy of Sciences 88.6 (1991): 2297-2301.
>
> [7] Richman, Joshua S., and J. Randall Moorman. "Physiological time-series analysis using approximate entropy and sample entropy." American journal of physiology-heart and circulatory physiology 278.6 (2000): H2039-H2049.
>
> [8] Krause, Andreas, and Daniel Golovin. "Submodular function maximization." Tractability 3.71-104 (2014): 3.

---

### Meta-Review · Area_Chair_RxSb · 2023-12-12

**Metareview:**

This paper was reviewed by five experts in the field. The authors' made some great improvements in the rebuttal and resolved most of the concerns. Reviewers liked the overall proposed method (hierarchical multimodal information flow) and strong experimental results on the datasets used in the paper. The major concerns are: 1) the proposed method heavily depends on the order of modality; 2) while the proposed method seems to be generic, the experiments focus on some narrow specific tasks.

The AC agrees with the reviewers' assessments and believes the paper provides an interesting method for multimodal learning. The proposed method is novel and uses a less popular paradigm yet achieved good results on some specific tasks. Although it's unclear whether the proposed method can be generalized to more popular multimodal learning tasks, the AC believes the strengths of the paper overweight the concerns. The authors are encouraged to make the necessary changes to the best of their ability to incorporate reviewers' suggestions and discussions during rebuttal.

**Justification For Why Not Higher Score:**

The proposed method is novel and interesting and showed some promising results on less popular tasks for multimodal learning. It's unclear whether the method can generalize and demonstrate strong performance on more general and popular multimodal tasks (e.g., VQA, image/video captioning, etc.).

**Justification For Why Not Lower Score:**

Most of the reviewers liked the paper. The concerns are mostly addressed during the rebuttal. While there are remaining concerns, the AC believes those concerns are minor compared to the strengths of the paper.

---

### Decision · Program_Chairs · 2024-01-16

Accept (poster)